# Aiming towards the minimizers: fast convergence of SGD for overparameterized problems

Chaoyue Liu[*], Dmitriy Drusvyatskiy[**], Yian Ma[*], Damek Davis[***], and Mikhail Belkin[*]

[*]Halicioğlu Data Science Institute, University of California San Diego
[**]Mathematics Department, University of Washington
[***]School of Operations Research and Information Engineering, Cornell University

## Abstract

Modern machine learning paradigms, such as deep learning, occur in or close to the interpolation regime, wherein the number of model parameters is much larger than the number of data samples. In this work, we propose a regularity condition within the interpolation regime which endows the stochastic gradient method with the same worst-case iteration complexity as the deterministic gradient method, while using only a single sampled gradient (or a minibatch) in each iteration. In contrast, all existing guarantees require the stochastic gradient method to take small steps, thereby resulting in a much slower linear rate of convergence. Finally, we demonstrate that our condition holds when training sufficiently wide feedforward neural networks with a linear output layer.

## 1 Introduction

Recent advances in machine learning and artificial intelligence have relied on fitting highly overparameterized models, notably deep neural networks, to observed data; e.g. [39, 37, 16, 23]. In such settings, the number of parameters of the model is much greater than the number of data samples, thereby resulting in models that achieve near-zero training error. Although classical learning paradigms caution against overfitting, recent work suggests ubiquity of the "double descent" phenomenon [3], wherein significant overparameterization actually improves generalization. The stochastic gradient method is the workhorse algorithm for fitting overparametrized models to observed data and understanding its performance is an active area of research. The goal of this paper is to obtain improved convergence guarantees for SGD in the interpolation regime that better align with its performance in practice.

Classical optimization literature emphasizes conditions akin to strong convexity as the phenomena underlying rapid convergence of numerical methods. In contrast, interpolation problems are almost never convex, even locally, around their solutions [24]. Furthermore, common optimization problems have complex symmetries, resulting in nonconvex sets of minimizers. Case in point, standard formulations for low-rank matrix recovery [4] are invariant under orthogonal transformations while ReLU neural networks are invariant under rebalancing of adjacent weight matrices [9]. The Polyak-Łojasiewicz (PŁ) inequality, introduced independently in [26] and [34], serves as an alternative to strong convexity that holds often in applications and underlies rapid convergence of numerical algorithms. Namely, it has been known since [34] that gradient descent convergences under the PŁ condition at a linear rate $O(\exp(-t/\kappa))$, where $\kappa$ is the condition number of the function.[1] In contrast, convergence guarantees for the stochastic gradient method under PŁ—the predominant algorithm in practice—are much less satisfactory. Indeed, all known results require SGD to take

---

[1]In particular, for a smooth function with $\beta$-Lipschitz gradient satisfying a PŁ inequality with constant $\alpha$, the condition number is $\kappa = \beta/\alpha$.

37th Conference on Neural Information Processing Systems (NeurIPS 2023).

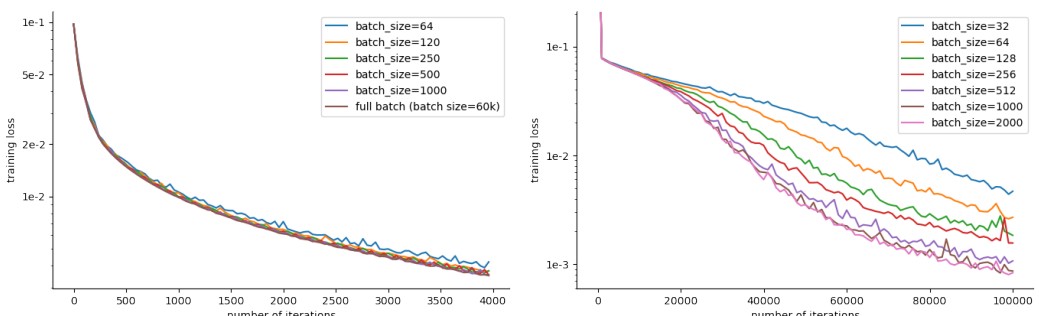

Figure 1: Convergence plot of SGD when training a fully connected neural network with 3 hidden layers and 1000 neurons in each on MNIST (left) and a ResNet-28 on CIFAR-10 (right). MNIST has 60k images and the convergence behavior stabilizes with small batchsize of m = 64; CIFAR-10 has 60k images and the convergence behavior stabilizes with batchsize of $m = 512$, although early on in training (first $20k$ iterations) convergence behavior looks identical for all batchsizes.

shorter steps than gradient descent to converge at all [2, 12], with the disparity between the two depending on the condition number $\kappa$. The use of the small stepsize directly translates into a slow rate of convergence. This requirement is in direct contrast to practice, where large step-sizes are routinely used. As a concrete illustration of the disparity between theory and practice, Figure 1 depicts the convergence behavior of SGD for training a neural network on the MNIST data set. As is evident from the Figure, even for small batch sizes, the linear rate of convergence of SGD is comparable to that of GD with an identical stepsize $\eta = 0.1$. Moreover, experimentally, we have verified that the interval of stepsizes leading to convergence for SGD is comparable to that of GD; indeed, the two are off only by a factor of 20. Using large stepsizes also has important consequences for generalization. Namely, recent works [19, 5] suggest that large stepsizes bias the iterates towards solutions that generalize better to unseen data. Therefore understanding the dynamics of SGD with large stepsizes is an important research direction. The contribution of our work is as follows.

> In this work, we highlight regularity conditions that endow SGD with a fast linear rate of convergence $\exp(-t/\kappa)$ both in expectation and with high probability, even when the conditions hold only locally. Moreover, we argue that the conditions we develop are reasonable because they provably hold on any compact region when training sufficiently wide feedforward neural networks with a linear output layer.

## 1.1 Outline of main results.

We will focus on the problem of minimizing a loss function $\mathcal{L}$ under the following two assumptions. First, we assume that $\mathcal{L}$ grows quadratically away from its set of global minimizers $S$:

$$\mathcal{L}(w) \geq \frac{\alpha}{2} \cdot \mathrm{dist}^2(w, S) \qquad \forall w \in B_r(w_0), \qquad \text{(QG)}$$

where $B_r(w_0)$ is a ball of radius $r$ around the initial point $w_0$. This condition is standard in the optimization literature and is implied for example by the PŁ-inequality holding on the ball $B_r(w_0)$; see Section A.2. Secondly, and most importantly, we assume there there exist constants $\theta, \rho > 0$ satisfying the aiming condition:[2]

$$\langle \nabla \mathcal{L}(w), w - \mathrm{proj}_S(w) \rangle \geq \theta \cdot \mathcal{L}(w) \qquad \forall w \in B_r(w_0). \qquad \text{(Aiming)}$$

Here, $\mathrm{proj}_S(w)$ denotes a nearest point in $S$ to $w$ and $\mathrm{dist}(w, S)$ denotes the distance from $w$ to $S$. The aiming condition ensures that the negative gradient $-\nabla \mathcal{L}(w)$ points towards $S$ in the sense that $-\nabla \mathcal{L}(w)$ correlated nontrivially with the direction $\mathrm{proj}_S(w) - w$. At first sight, the aiming condition appears similar to quasar-convexity, introduced in [14] and further studied in [15, 22, 20]. Namely a function $\mathcal{L}$ is *quasar-convex* relative to a fixed point $\bar{w} \in S$ if the estimate (Aiming) holds with $\mathrm{proj}_S(w)$ replaced by $\bar{w}$. Although the distinction between aiming and quasar-convexity may

---

[2]If $\mathrm{proj}_S(w)$ is not a singleton, $\mathrm{proj}_S(w)$ in the expression should be replaced with any element $\bar{w}$ from $\mathrm{proj}_S(w)$.

appear mild, it is significant. As a concrete example, consider the function $\mathcal{L}(x, y) = \frac{1}{2}(y - ax^2)^2$ for any $a > 0$. It is straightforward to see that $\mathcal{L}$ satisfies the aiming condition on some neighborhood of the origin. However, for any neighborhood $U$ of the origin, the function $\mathcal{L}$ is not quasar-convex on $U$ relative to any point $(x, ax^2) \in U$; see Section C. More generally, we show that (Aiming) holds automatically for any $C^3$ smooth function $\mathcal{L}$ satisfying (QG) locally around the solution set. Indeed, we may shrink the neighborhood to ensure that $\theta$ is arbitrarily close to 2. Secondly, we show that (Aiming) holds for sufficiently wide feedforward neural networks with a linear output layer.

Our first main result can be summarized as follows. Roughly speaking, as long as the SGD iterates remain in $B_r(w_0)$, they converge to $S$ at a fast linear rate $O(\exp(-t\alpha\theta^2/\beta))$ with high probability.

**Theorem 1.1** (Informal). *Consider minimizing the function $\mathcal{L}(w) = \mathbb{E}\ell(w, z)$, where the losses $\ell(\cdot, z)$ are nonnegative and have $\beta$-Lipschitz gradients. Suppose that the minimal value of $\mathcal{L}$ is zero and both regularity conditions* (QG) *and* (Aiming) *hold. Then as long as the SGD iterates $w_t$ remain in $B_r(w_0)$, they converge to $S$ at a linear rate $O(\exp(-t\alpha\theta^2/\beta))$ with high probability.*

The proof of the theorem is short and elementary. The downside is that the conclusion of the theorem is conditional on the iterates remaining in $B_r(w_0)$. Ideally, one would like to estimate this probability as a function of the problem parameters. With this in mind, we show that for a special class of nonlinear least squares problems, including those arising when fitting wide neural networks, this probability may be estimated explicitly. The end result is the following unconditional theorem.

**Theorem 1.2** (Informal). *Consider the loss $\mathcal{L}(w) = \frac{1}{n}\sum_{i=1}^{n}(f(w, x_i) - y_i)^2$, where $f(w, \cdot)$ is a fully connected neural network with $l$ hidden layers and a linear output layer. Let $\lambda_\infty$ be the minimal eigenvalue of the Neural Tangent Kernel of an infinitely wide neural network. Then with high probability both conditions* (QG) *and* (Aiming) *hold on a ball of radius $r$ around the initial point $w_0 \sim N(0, I)$ with $\theta = 1$ and $\alpha = \lambda_\infty/2$, as long as the network width $m$ satisfies $m = \tilde{\Omega}(nr^{6l+2}/\lambda_\infty^2)$. If in addition $r = \Omega(1/\delta\sqrt{\lambda_\infty})$, with probability $1 - \delta$, SGD with stepsize $\eta = \Theta(1)$ converges to a zero-loss solution at the fast rate $O(\exp(-t\lambda_\infty))$. This parameter regime is identical as for gradient descent to converge in [24], with the only exception of the inflation of $r$ by $1/\delta$.*

A key part of the argument is to estimate the probability that the iterates remain in a ball $B_r(w_0)$. A naive approach is to bound the length of the iterate trajectory in expectation, but this would then require the radius $r$ to expand by an additional factor of $1/\lambda_\infty$, which in turn would increase $m$ multiplicatively by $\lambda_\infty^{-6l-2}$. We avoid this exponential blowup by a careful stopping time argument and the transition to linearity phenomenon that has been shown to hold for sufficiently wide neural networks [24]. While Theorem 1.2 is stated with a constant failure probability, there are standard ways to remove the dependence. One option is to simply set $\delta_1 = \Omega(1)$ and rerun SGD logarithmically many times from the the same initialization $w_0$ and return the final iterate with smallest function value. Section 4 outlines a more nuanced strategy based on a small ball assumption, which entirely avoids computation of the function values of the full objective.

## 1.2 Comparison to existing work.

We next discuss how our results fit within the existing literature, summarized in Table 1. Setting the stage, consider the problem of minimizing a smooth function $\mathcal{L}(w) = \mathbb{E}\ell(w, z)$ and suppose for simplicity that its minimal value is zero. We say that $\mathcal{L}$ satisfies the *Polyak-Łojasiewicz (PŁ) inequality* if there exists $\alpha > 0$ satisfying

$$\|\nabla\mathcal{L}(w)\|^2 \geq 2\alpha \cdot \mathcal{L}(w), \tag{PŁ}$$

for all $w \in \mathbf{R}^d$. In words, the gradient $\nabla\mathcal{L}(w)$ dominates the function value $\mathcal{L}(w)$, up to a power. Geometrically, such functions have the distinctive property that the gradients of the rescaled function $\sqrt{\mathcal{L}(w)}$ are uniformly bounded away from zero outside the solution set. See Figures 2a and 2b for an illustration. Using the PŁ inequality, we may associate to $\mathcal{L}$ two condition numbers, corresponding to the full objective and its samples, respectively. Namely, we define $\bar{\kappa} \triangleq \bar{\beta}/\alpha$ and $\kappa \triangleq \beta/\alpha$, where $\bar{\beta}$ is a Lipschitz constant of the full gradient $\nabla\mathcal{L}$ and $\beta$ is a Lipschitz constant of the sampled gradients $\nabla\ell(\cdot, z)$ for all $z$. Clearly, the inequality $\bar{\kappa} \leq \kappa$ holds and we will primarily be interested in settings where the two are comparable.

The primary reason why the PŁ condition is useful for optimization is that it ensures linear convergence of gradient-type algorithms. Namely, it has been known since Polyak's seminal work

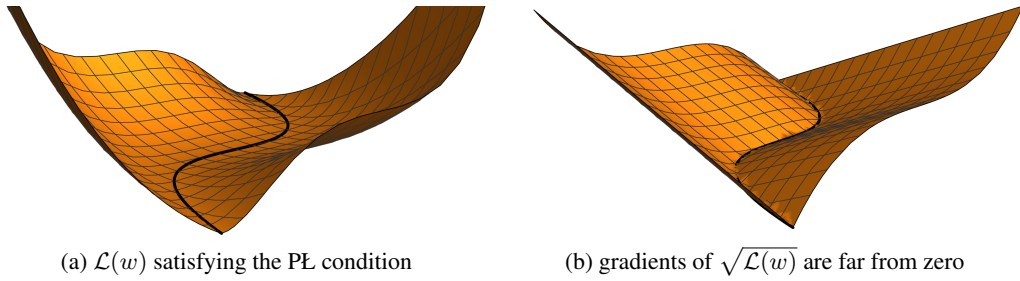

(a) $\mathcal{L}(w)$ satisfying the PŁ condition      (b) gradients of $\sqrt{\mathcal{L}(w)}$ are far from zero

[34] that the full-batch gradient descent iterates $w_{t+1} = w_t - \frac{1}{\beta}\nabla\mathcal{L}(w_t)$ converge at the linear rate $O(\exp(-t/\bar{\kappa}))$. More recent papers have extended results of this type to a wide variety of algorithms both for smooth and nonsmooth optimization [27, 7, 21, 30, 1] and to settings when the PŁ inequality holds only locally on a ball [24, 32].

For stochastic optimization problems under the PŁ condition, the story is more subtle, since the rates achieved depend on moment bounds on the gradient estimator, such as:

$$\mathbb{E}[\|\nabla\ell(w, z)\|^2] \le A\mathcal{L}(w) + B\|\nabla\mathcal{L}(w)\|^2 + C, \tag{1.1}$$

for $A, B, C \ge 0$. In the setting where $C > 0$—the classical regime– stochastic gradient methods converge sublinearly at best, due to well-known lower complexity bounds in stochastic optimization [31]. On the other hand, in the setting where $C = 0$—interpolation problems—stochastic gradient methods converge linearly when equipped with an appropriate stepsize, as shown in [2, Theorem 1], [22, Corollary 2], [38], and [13, Theorem 4.6]. Although linear convergence is assured, the rate of linear converge under the PŁ condition and interpolation is an order of magnitude worse than in the deterministic setting. Namely, the three papers[2, Theorem 1], [22, Corollary 2] and [13, Theorem 4.6] obtain linear rates on the order of $\exp(-t/\bar{\kappa}\kappa)$. On the other hand, in the case $A = C = 0$, which is called the strong growth property, the paper [38, Theorem 4] yields the seemingly better rate $\exp(-t/B\bar{\kappa})$. The issue, however, is that $B$ can be extremely large. As an illustrative example, consider the loss functions $\ell(w, z) = \frac{1}{2}\text{dist}^2(w, Q_z)$ where $Q_z$ are smooth manifolds. A quick computation shows that equality $\|\nabla\ell(w, z)\|^2 = 2\ell(w, z)$ holds. Therefore, locally around the intersection of $\cap_z Q_z$, the estimate (1.1) with $A = C = 0$ is *exactly equivalent* to the PŁ-condition with $B = 1/\alpha$. As a further illustration, Figure 3 shows the possible large value of the constant $B$ along the SGD iterates for training a neural network on MNIST. Another related paper is [36]: assuming so-called small gradient confusion and requiring a stronger version of PL condition (i.e., each individual loss $l_i$ is $\mu$-PL), the paper [36] showed a slow rate of convergence $\exp(-t/n^2\kappa)$, where $n$ is the dataset size.

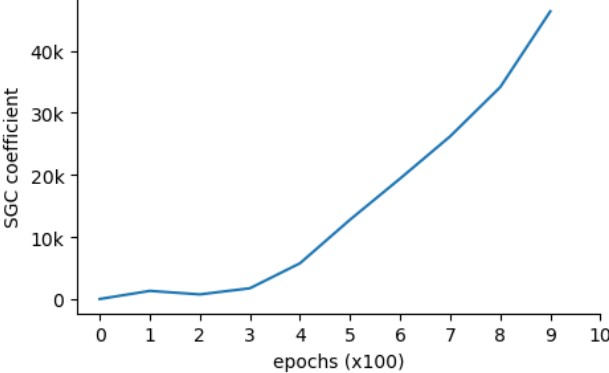

Figure 3: We train a fully-connected neural network on the MNIST dataset. The network has 4 hidden layers, each with 1024 neurons. We optimize the MSE loss using SGD with a batch size 512 and a learning rate 0.5. The training was run over 1k epochs, and the ratio $\mathbb{E}[\|\nabla\ell(w, z)\|^2]/\|\nabla\mathcal{L}(w)\|^2$ is evaluated every 100 epochs. The ratio grows almost linearly during training, suggesting that strong growth is practically not satisfied with a constant coefficient $B$.

| Reference | Bound on $\mathbb{E}[\|\nabla\ell(w,z)\|^2]$ | Quasar Convex? | Rate |
|---|---|---|---|
| [2, Theorem 1] | $2\beta(\mathcal{L}(w) - \mathcal{L}^*)$ | No | $\exp\left(-\frac{t}{\kappa\bar{\kappa}}\right)$ |
| [38, Theorem 4] | $B\|\nabla\mathcal{L}(w)\|^2$ | No | $\exp\left(-\frac{t}{B\bar{\kappa}}\right)$ |
| [22, Corollary 2] | $A\mathcal{L}(w) + B\|\nabla\mathcal{L}(w)\|^2$ | No | $\exp\left(-\frac{t}{\bar{\kappa}\max\{B,A/\alpha\}}\right)$ |
| [13, Theorem 4.6] | $2\beta(\mathcal{L}(w) - \mathcal{L}^*) + \|\nabla\mathcal{L}(w)\|^2$ | No | $\exp\left(\frac{-t}{\kappa\bar{\kappa}}\right)$ |
| [11, Corollary 3.3] | $\sigma^2 + 2\alpha\|w - w_\star\|^2$ | Yes | Sublinear |
| [20, Theorem 4.4] | $\sigma^2 + \|\nabla\mathcal{L}(w)\|^2$ | Yes | Sublinear |
| **This work** | $2\beta(\mathcal{L}(w) - \mathcal{L}^*)$ | (Aiming) | $\exp\left(\frac{-t\theta^2}{\kappa}\right)$ |

Table 1: Comparison to recent work on nonconvex stochastic gradient methods under the $\alpha$-PŁ and smoothness conditions. We define $\bar{\kappa} = \bar{\beta}/\alpha$ and $\kappa = \beta/\alpha$, where $\bar{\beta}$ is a Lipschitz constant of the full gradient $\nabla\mathcal{L}$ and $\beta$ is a Lipschitz constant of the sampled gradients $\nabla\ell(\cdot,z)$ for all $z$.

The purpose of this work is to understand whether we can improve stepsize selection and the convergence rate of SGD for nonconvex problems under the (local) PŁ condition and interpolation. Unfortunately, the PŁ condition alone appears too weak to yield improved rates. Instead, we take inspiration from recent work on accelerated deterministic nonconvex optimization, where the recently introduced quasar-convexity condition has led to improved rates [15]. We note that quasar-convexity is a very restrictive assumption in the interpolation regime because it requires the solution set to be an affine subspace; see the discussion in Appendix E. Recent work has also shown that quasar convexity can lead to accelerated *sublinear* rates of convergence for certain stochastic optimization problems [20, Theorem 4.4] (and the concurrent work [11, Corollary 3.3]), but to the best of our knowledge, there are no works that analyze improved linear rates of convergence. Thus, in this work, we fill the gap in the literature, by providing a rate that matches that of deterministic gradient descent and allows for a large stepsize. Moreover, in contrast to most available results, we only assume that regularity conditions hold on a ball—the common setting in applications. The local nature of the assumptions requires us to bound the probability of the iterates escaping.

## 2 Main results

Throughout the paper, we will consider the stochastic optimization problem

$$\min_w \mathcal{L}(w) \triangleq \mathop{\mathbb{E}}_{z\sim\mathcal{P}} \ell(w,z),$$

where $\mathcal{P}$ is a probability distribution that is accessible only through sampling and $\ell(\cdot,z)$ is a differentiable function on $\mathbf{R}^d$. We let $S$ denote the set of minimizers of $\mathcal{L}$. We impose that $\mathcal{L}$ satisfies the following assumptions on a set $\mathcal{W}$. The two main examples are when $\mathcal{W}$ is a ball $B_r(w_0)$ and when $\mathcal{W}$ is a tube around the solution set:

$$S_r \triangleq \{w \in \mathbf{R}^d : \text{dist}(w,S) \leq r\}.$$

**Assumption 1** (Running assumptions). Suppose that there exist constants $\alpha,\beta,\theta \geq 0$ and a set $\mathcal{W} \subset \mathbf{R}^d$ satisfying the following.

1. **(Interpolation)** The losses $\ell(w,z)$ are nonnegative, the minimal value of $\mathcal{L}$ is zero, and the set of minimizers $S \triangleq \text{argmin}\,\mathcal{L}$ is nonempty.

2. **(Smoothness)** For almost every $z \sim \mathcal{P}$, the loss $\ell(\cdot,z)$ is differentiable and the gradient $\nabla\ell(\cdot,z)$ is $\beta$-Lipschitz continuous on $\mathcal{W}$.

3. **(Quadratic growth)** The estimate holds:

$$\mathcal{L}(w) \geq \frac{\alpha}{2} \cdot \text{dist}^2(w,S) \qquad \forall w \in \mathcal{W}. \tag{2.1}$$

4. **(Aiming)** For all $w \in \mathcal{W}$ there exists a point $\bar{w} \in \text{proj}(w,S)$ such that

$$\langle \nabla\mathcal{L}(w), w - \bar{w} \rangle \geq \theta \cdot \mathcal{L}(w). \tag{2.2}$$

We define the condition number $\kappa \triangleq \beta/\alpha$.

As explained in the introduction, the first three conditions (1)-(3) are classical in the literature. In particular, both quadratic growth (3) on a ball $\mathcal{W} = B_r(w_0)$ and existence of solutions in $\mathcal{W}$ follow from a local PŁ-inequality. In order to emphasize the local nature of the condition, following [24] we say that $\mathcal{L}$ is $\alpha$-$PŁ^*$ on $\mathcal{W}$ if the inequality (PŁ) holds for all $w \in \mathcal{W}$. We recall the proof of the following lemma in Section A.2.

**Lemma 2.1** (PŁ$^*$ condition implies quadratic growth). *Suppose that $\mathcal{L}$ is differentiable and is $\alpha$-$PŁ^*$ on a ball $B_{2r}(w_0)$. Then as long as $\mathcal{L}(w_0) < \frac{1}{2}\alpha r^2$, the intersection $S \cap B_r(w_0)$ is nonempty and*

$$\mathcal{L}(w) \geq \frac{\alpha}{8}\mathrm{dist}^2(w, S) \qquad \forall w \in B_r(w_0).$$

The aiming condition (4) is very closely related to *quasar-convexity*, which requires (2.2) to hold for all $w \in \mathbf{R}^d$ and a *distinguished point* $\bar{w} \in S$ that is independent of $w$. This distinction may seem mild, but is in fact important because aiming holds for a much wider class of problems. As a concrete example, consider the function $\mathcal{L}(x, y) = \frac{1}{2}(y - ax^2)^2$ for any $a > 0$. It is straightforward to see that $\mathcal{L}$ satisfies the aiming condition on some neighborhood of the origin. However, for any neighborhood $U$ of the origin, the function $\mathcal{L}$ is not quasar-convex on $U$ relative to any point $(x, ax^2) \in U$; see Section C. We now show that (4) is valid locally for any $C^3$-smooth function satisfying quadratic growth, and we may take $\theta$ arbitrarily close to 2 by shrinking $r$. Later, we will also show that problems of learning wide neural networks also satisfy the aiming condition.

**Theorem 2.2** (Local aiming). *Suppose that $\mathcal{L}$ is $C^2$-smooth and $\nabla^2\mathcal{L}$ is $L$-Lipschitz continuous on the tube $S_r$. Suppose moreover that $\mathcal{L}$ satisfies the quadratic growth condition (2.1) and $r < \frac{6\alpha}{5L}$. Then the aiming condition (2.2) holds with parameter $\theta = 2 - \frac{5Lr}{3\alpha}$. An analogous statement holds if $S_r$ is replaced by a ball $B_r(\bar{w}_0)$ for some $\bar{w}_0 \in S$.*

---

**Algorithm 1** SGD$(w_0, \eta, T)$

---

**Initialize:** Initial $w_0 \in \mathbf{R}^d$, learning rate $\eta > 0$, iteration counter $T \in \mathbb{N}$.
**For** $t = 1, \ldots, T - 1$ **do:**

$$\text{Sample } z_t \sim \mathcal{P}$$
$$\text{Set } w_{t+1} = w_t - \eta \nabla \ell(w_t, z_t).$$

**Return:** $w_T$.

---

## 2.1 SGD under regularity on a tube $S_r$

Convergence analysis for SGD (Algorithm 1) is short and elementary in the case $\mathcal{W} = S_r$ and therefore this is where we begin. We note, however, that the setting $\mathcal{W} = B_r(w_0)$ is much more realistic, as we will see, but also more challenging.

The converge analysis of SGD proceeds by a familiar one-step contraction argument.

**Lemma 2.3** (One-step contraction on a tube). *Suppose that Assumption 1 holds on a tube $\mathcal{W} = S_{2r}$ and fix a point $w \in S_r$. Define the updated point $w^+ = w - \eta \nabla f(w, z)$ where $z \sim \mathcal{P}$. Then for any stepsize $\eta < \frac{\theta}{\beta}$, the estimate holds:*

$$\mathbb{E}_{z \sim \mathcal{P}} \mathrm{dist}^2(w^+, S) \leq (1 - \alpha\eta(\theta - \beta\eta)) \mathrm{dist}^2(w, S). \qquad (2.3)$$

Using the one step guarantee of Lemma 2.3, we can show that SGD iterates converge linearly to $S$ if Assumption 1 holds on a tube $\mathcal{W} = S_{2r}$. The only complication is to argue that the iterates are unlikely to leave the tube if we start in a slightly smaller tube $S_{r'}$ for some $r' < r$. We do so with a simple stopping time argument.

**Theorem 2.4** (Convergence on a tube). *Suppose that Assumption 1 holds relative to a tube $\mathcal{W} = S_{2r}$ for some constant $r > 0$. Fix a stepsize $\eta > 0$ satisfying $\eta < \frac{\theta}{\beta}$. Fix a constant $\delta_1 > 0$ and a point*

$w_0 \in S_{\sqrt{\delta_1} r}$. Then with probability at least $1 - \delta_1$, the SGD iterates $\{w_t\}_{t \geq 0}$ remain in $\mathcal{W}$. Moreover, with probability at least $1 - \delta_1 - \delta_2$, the estimate $\operatorname{dist}^2(w_t, S) \leq \varepsilon \cdot \operatorname{dist}^2(w_0, S)$ holds after

$$ t \geq \frac{1}{\alpha \eta (\theta - \beta \eta)} \log \left( \frac{1}{\delta_2 \varepsilon} \right) \qquad \text{iterations.} $$

Thus as long SGD is initialized at a point $w_0 \in S_{\sqrt{\delta_1} r}$, with probability at least $1 - \delta_1 - \delta_2$, the iterates remain in $S_r$ and converge at linear rate $O(\frac{1}{\delta_2} \exp(-t\theta^2 \alpha/\beta))$. Note that the dependence on $\delta_2$ is logarithmic, while the dependence on $\delta_1$ appears linearly in the initialization requirement $w_0 \in S_{\sqrt{\delta_1} r}$. One simple way to remove the dependence on $\delta_1$ is to simply rerun the algorithm from the same initial point logarithmically many times and return the point with the smallest function value. An alternative strategy that bypasses evaluating function values will be discussed in Section 4.

## 2.2 SGD under regularity on a ball $B_r(w_0)$

Next, we describe convergence guarantees for SGD when Assumption 1 holds on a ball $\mathcal{W} = B_r(w_0)$. The key complication is the following. While $w_t$ are in the ball, the distance $\operatorname{dist}^2(w_t, S)$ shrinks in expectation. However, the iterates may in principle quickly escape the ball $B_r(w_0)$, after which point we lose control on their progress. Thus we must lower bound the probability that the iterates $w_t$ remain in the ball. To this end, we will require the following additional assumption.

**Assumption 2** (Uniform aiming). The estimate

$$ \langle \nabla \mathcal{L}(w), w - v \rangle \geq \theta \mathcal{L}(w) - \rho \cdot \operatorname{dist}(w, S) \tag{2.4} $$

holds for all $w \in B_r(w_0)$ and $v \in B_r(w_0) \cap S$.

The intuition underlying this assumption is as follows. We would like to replace $\bar{w}$ in the aiming condition (2.2) by an arbitrary point $v \in B_r(w_0) \cap S$, thereby having a condition of the form $\langle \nabla \mathcal{L}(w), w - v \rangle \geq \theta \cdot \mathcal{L}(w)$. The difficulty is that this condition may not be true for the main problem we are interested in— training wide neural networks. Instead, it suffices to lower bound the inner product by $\theta \mathcal{L}(w) - \rho \cdot \operatorname{dist}(w, S)$ where $\rho$ is a small constant. This weak condition provably holds for wide neural networks, as we will see in the next section. The following is our main result.

**Theorem 2.5** (Convergence on a ball). *Suppose that Assumptions 1 and 2 hold on a ball $\mathcal{W} = B_{3r}(w_0)$. Fix constants $\delta_1 \in (0, \frac{1}{3})$ and $\delta_2 \in (0, 1)$, and assume $\operatorname{dist}^2(w_0, S) \leq \delta_1^2 r^2$. Fix a stepsize $\eta < \frac{\theta}{\beta}$ and suppose $\rho \leq (\theta - \beta \eta) \alpha r$. Then with probability at least $1 - 5\delta_1$, all the SGD iterates $\{w_t\}_{t \geq 0}$ remain in $B_r(w_0)$. Moreover, with probability at least $1 - 5\delta_1 - \delta_2$, the estimate $\operatorname{dist}^2(w_t, S) \leq \varepsilon \cdot \operatorname{dist}^2(w_0, S)$ holds after*

$$ t \geq \frac{1}{\alpha \eta (\theta - \beta \eta)} \log \left( \frac{1}{\varepsilon \delta_2} \right) \qquad \text{iterations.} $$

Thus as long as $\rho$ is sufficiently small and the initial distance satisfies $\operatorname{dist}^2(w_0, S) \leq \delta_1^2 r^2$, with probability at least $1 - 5\delta_1 - \delta_2$, the iterates remain in $B_r(w_0)$ and converge at a fast linear rate $O(\frac{1}{\delta_2} \exp(-t\theta^2 \alpha/\beta))$. While the dependence on $\delta_2$ is logarithmic, the constant $\delta_1$ linearly impacts the initialization region. Section 4 discusses a way to remove this dependence. As explained in Lemma 2.1, both quadratic growth and the initialization quality holds if $\mathcal{L}$ is $\alpha$-PŁ$^*$ on the ball $B_r(w_0)$, and $r$ is sufficiently big relative to $1/\alpha$.

# 3 Consequences for nonlinear least squares and wide neural networks

We next discuss the consequences of the results in the previous sections to nonlinear least squares and training of wide neural networks. To this end, we begin by verifying the aiming (2.2) and uniform aiming (2.4) conditions for nonlinear least squares. The key assumption we will make is that the nonlinear map's Jacobian $\nabla F$ has a small Lipschitz constant in operator norm.

**Theorem 3.1.** *Consider a function $\mathcal{L}(w) = \frac{1}{2} \|F(w)\|^2$, where $F : \mathbf{R}^d \to \mathbf{R}^n$ is $C^1$-smooth. Suppose that there is a point $w_0$ satisfying $\operatorname{dist}(w_0, S) \leq r$ and such that on the ball $B_{2r}(w_0)$, the gradient $\nabla \mathcal{L}$ is $\beta$-Lipschitz, the Jacobian $\nabla F$ is $L$-Lipschitz in the operator norm, and the quadratic growth condition (2.1) holds. Then as long as $L \leq \frac{2\alpha}{r\sqrt{\beta}}$, the aiming (2.2) and uniform aiming (2.4) conditions hold on $B_r(w_0)$ with $\theta = 2 - \frac{rL\sqrt{\beta}}{\alpha}$ and $\rho = 8r^2 L\sqrt{\beta}$.*

We next instantiate Theorem 3.1 and Theorem 2.5 for a nonlinear least squares problem arising from fitting a wide neural network. Setting the stage, an $l$-layer (feedforward) neural network $f(w; x)$, with parameters $w$, input $x$, and linear output layer is defined as follows:

$$\alpha^{(0)} = x,$$
$$\alpha^{(i)} = \sigma\left(\frac{1}{\sqrt{m_{i-1}}} W^{(i)} \alpha^{(i-1)}\right), \quad \forall i = 1, \ldots, l-1$$
$$f(w; x) = \frac{1}{\sqrt{m_{l-1}}} W^{(l)} \alpha^{(l-1)}.$$

Here, $m_i$ is the width (i.e., number of neurons) of $i$-th layer, $\alpha^{(i)} \in \mathbf{R}^{m_i}$ denotes the vector of $i$-th hidden layer neurons, $w := \{W^{(1)}, W^{(2)}, \ldots, W^{(l)}, W^{(l+1)}\}$ denotes the collection of the parameters (or weights) $W^{(i)} \in \mathbb{R}^{m_i \times m_{i-1}}$ of each layer, and $\sigma$ is the activation function, e.g., $sigmoid$, $tanh$, linear activation. We also denote the width of the neural network as $m := \min_{i \in [l]} m_i$, i.e., the minimal width of the hidden layers. The neural network is usually randomly initialized, i.e., each individual parameter is initialized i.i.d. following $\mathcal{N}(0, 1)$. Henceforth, we assume that the activation functions $\sigma$ are twice differentiable, $L_\sigma$-Lipschitz, and $\beta_\sigma$-smooth. In what follows, the order notation $\Omega(\cdot)$ and $O(\cdot)$ will suppress multiplicative factors of polynomials (up to degree $l$) of the constants $C$, $L_\sigma$ and $\beta_\sigma$.

Given a dataset $\mathcal{D} = \{(x_i, y_i)\}_{i=1}^n$, we fit the neural network by solving the least squares problem

$$\min_w \; \mathcal{L}(w) \triangleq \tfrac{1}{2}\|F(w)\|^2 \qquad \text{where} \qquad \tfrac{1}{2}\|F(w)\|^2 = \frac{1}{n}\sum_{i=1}^n (f(w, x_i) - y_i)^2.$$

We assume that all the the data inputs $x_i$ are bounded, i.e., $\|x_i\| \leq C$ for some constant $C$.

Our immediate goal is to verify the assumptions of Theorem 3.1, which are quadratic growth and (uniform) aiming. We begin with the former. Quadratic growth is a consequence of the PŁ-condition. Namely, define the Neural Tangent Kernel $K(w_0) = \nabla F(w_0)\nabla F(w_0)^\top$ at the random initial point $w_0 \sim N(0, I)$ and let $\lambda_0$ be the minimal eigenvalue of $K(w_0)$. The value $\lambda_0$ has been shown to be positive with high probability in [10, 8]. Specifically, it was shown that, under a mild non-degeneracy condition on the data set, the smallest eigenvalue $\lambda_\infty$ of NTK of an infinitely wide neural network is positive (see Theorem 3.1 of [10]). Moreover, if the network width satisfies $m = \Omega(\frac{n^2 \cdot 2^{O(l)}}{\lambda_\infty^2} \log \frac{nl}{\epsilon})$, then with probability at least $1 - \epsilon$ the estimate $\lambda_0 > \frac{\lambda_\infty}{2}$ holds [8, Remark E.7]. Of course, this is worst case bound and for our purposes we will only need to ensure that $\lambda_0$ is positive. It will also be important to know that $\|F(w_0)\|^2 = O(1)$, which indeed occurs with high probability as shown in [18]. To simplify notation, let us lump these two probabilities together and define

$$p \triangleq \mathbb{P}\{\lambda_0 > 0, \|F(w_0)\|^2 \leq C\}.$$

Next, we require the following theorem, which shows two fundamental properties on $B_r(w_0)$ when the width $m$ is sufficiently large: (1) the function $w \mapsto f(w, x)$ is nearly linear and (2) the function $\mathcal{L}$ satisfies the PŁ condition with parameter $\lambda_0/2$.

**Theorem 3.2** (Transition to linearity [25] and the PŁ condition [24]). *Given any radius $r > 0$, with probability $1 - p - 2\exp(-\frac{ml}{2}) - (1/m)^{\Theta(\ln m)}$ of initialization $w_0 \sim N(0, I)$, it holds:*

$$\|\nabla^2 f(w, x)\|_{\mathrm{op}} = \tilde{O}\left(\frac{r^{3l}}{\sqrt{m}}\right) \qquad \forall w \in B_r(w_0), \|x\| \leq C. \tag{3.1}$$

*In the same event, as long as the width of the network satisfies $m = \tilde{\Omega}\left(\frac{nr^{6l+2}}{\lambda_0^2}\right)$, the function $\mathcal{L}$ is PŁ\* on $B_r(w_0)$ with parameter $\lambda_0/2$.*

Note that (3.1) directly implies that the Lipschitz constant of $\nabla F$ is bounded by $\tilde{O}\left(\frac{r^{3l}}{\sqrt{m}}\right)$ on $B_r(w_0)$, and can therefore be made arbitrarily small. Quadratic growth is now a direct consequence of the PŁ\* condition while (uniform) aiming follows from an application of Theorem 3.1.

**Theorem 3.3** (Aiming and quadratic growth condition for wide neural network). *With probability at least $1 - p - 2\exp(-\frac{ml}{2}) - (1/m)^{\Theta(\ln m)}$ with respect to the initialization $w_0 \sim N(0, I)$, as long as*

$$m = \tilde{\Omega}\left(\frac{nr^{6l+2}}{\lambda_0^2}\right) \qquad \text{and} \qquad r = \Omega\left(\frac{1}{\sqrt{\lambda_0}}\right),$$

*the following are true:*

1. *the quadratic growth condition* (2.1) *holds on $B_r(w_0)$ with parameter $\lambda_0/2$ and the intersection $B_r(w_0) \cap S$ is nonempty,*

2. *aiming* (2.2) *and uniform aiming* (2.4) *conditions hold in $B_r(w_0)$ with $\theta = 1$ and $\rho = \tilde{O}\left(\frac{r^{3l+2}}{\sqrt{m}}\right)$,*

3. *the gradient of each function $\ell_i(w) \triangleq (f(w, x_i) - y_i)^2$ is $\beta$-Lipschitz on $B_r(w_0)$ with $\beta = O(1)$.*

Please see Appendix D for a numerical verification of an estimate of the aiming condition. It remains to deduce convergence guarantees for SGD by applying Theorem 2.5.

**Corollary 3.4** (Convergence of SGD for wide neural network)**.** *Fix constants $\delta_1 \in (0, \frac{1}{3})$, $\delta_2 \in (0, 1)$, $\varepsilon > 0$ and $t \in \mathbb{N}$. There is a stepsize $\eta = \Theta(1)$ such that the following is true. With probability at least $1 - p - \delta_1 - \delta_2 - 2\exp(-\frac{ml}{2}) - (1/m)^{\Theta(\ln m)}$, as long as*

$$m = \tilde{\Omega}\left(\frac{nr^{6l+2}}{\lambda_0^2}\right) \qquad and \qquad r = \Omega\left(\frac{1}{\delta_1 \sqrt{\lambda_0}}\right),$$

*all the SGD iterates $\{w_t\}_{t \geq 0}$ remain in $B_r(w_0)$ and the estimate $\mathrm{dist}^2(w_t, S) \leq \varepsilon \cdot \mathrm{dist}^2(w_0, S)$ holds after $t \geq \frac{1}{\lambda_0} \log\left(\frac{1}{\varepsilon \delta_2}\right)$ iterations.*

Thus, the width requirements for SGD to converge at a fast linear rate are nearly identical to those for gradient descent [24], with the exception being that the requirement $r = \Omega\left(\frac{1}{\sqrt{\lambda_0}}\right)$ is strengthened to $r = \Omega\left(\frac{1}{\delta_1 \sqrt{\lambda_0}}\right)$. That is, the radius $r$ needs to shrink by the probability of failure.

## 4 Boosting to high probability

A possible unsatisfying feature of Theorems 2.4 and 2.5 and Corollary 3.4 is that the size of the initialization region shrinks with the probability of failure $\delta_1$. A natural question is whether this requirement may be dropped. Indeed, we will now see how to boost the probability of success to be independent of $\delta_1$. A first reasonable idea is to simply rerun SGD a few times from an initialization region corresponding to $\delta_1 = 1/2$. Then by Hoeffding's inequality, after very trials, at least a third of them will be successful. The difficulty is to determine which trial was indeed successful. The fact that the solution set is not a singleton rules out strategies based on the geometric median of means [31, p. 243], [29]. Instead, we may try to estimate the function value at each of the returned points. In a classical setting of stochastic optimization, this is a very bad idea because it amounts to mean estimation, which in turn requires $O(1/\varepsilon^2)$ samples. The saving grace in the interpolation regime is that $\ell(w, \cdot)$ *is a nonnegative function of the samples*. While estimating the mean of nonnegative random variables still requires $O(1/\varepsilon^2)$ samples, detecting that a nonnegative random variable is large requires very few samples! This basic idea is often called the small ball principle and is the basis for establishing generalization bounds with heavy tailed data [28]. With this in mind, we will require the following mild condition, stipulating that the empirical average $\frac{1}{m}\sum_{i=1}^{m} \ell(w, z_i)$ to be lower bounded by $\mathcal{L}(w)$ with high probability over the iid samples $z_i$.

**Assumption 3** (Detecting large values)**.** *Suppose that there exist constants $c_1 > 0$ and $c_2 > 0$ such that for any $w \in \mathbf{R}^d$, integer $m \in \mathbb{N}$, and iid samples $z_1, \ldots, z_m \sim \mathcal{P}$, the estimate holds:*

$$P\left(\frac{1}{m}\sum_{i=1}^{m} \ell(w, z_i) \geq c_1 \mathcal{L}(w)\right) \geq 1 - \exp(-c_2 m).$$

Importantly, this condition does not have anything to do with light tails. A standard sufficient condition for Assumption 3 is a small ball property.

**Assumption 4** (Small ball)**.** *There exist constants $\tau > 0$ and $p \in (0, 1)$ satisfying*

$$\mathbb{P}\big(\ell(w, z) \geq \tau \cdot \mathcal{L}(w)\big) \geq p \qquad \forall w \in \mathbf{R}^d.$$

The small ball property simply asserts that $\ell(w, \cdot)$ should not put too much mess on small values relative to its mean $\mathcal{L}(w)$. Bernstein's inequality directly shows that Assumption 4 implies Assumption 3. We summarize this observation in the following theorem.

**Lemma 4.1.** *Assumption 4 implies Assumption 3 with $c_1 = \frac{p\tau}{2}$ and $c_2 = \frac{p}{4}$.*

A valid bound for the small ball probabilities is furnished by the Paley-Zygmund inequality [33]:

$$\mathbb{P}\big(\ell(w, z) \geq \tau \mathcal{L}(w)\big) \geq (1 - \tau)^2 \frac{\mathcal{L}(w)^2}{[\mathbb{E}\ell(w, z)^2]} \qquad \forall \tau \in [0, 1].$$

Thus if the ratio $\frac{\mathbb{E}\ell(w,z)^2}{\mathcal{L}(w)}$ is bounded by some $D > 0$, then the small ball condition holds with $p = (1 - \tau)^2/D$ where $\tau \in [0, 1]$ is arbitrary.

The following lemma shows that under Assumption 3, we may turn any estimation procedure for finding a minimizer of $\mathcal{L}$ that succeeds with constant probability into one that succeeds with high probability. The procedure simply draws a small batch of samples $z_1, \dots, z_m \sim \mathcal{P}$ and rejects those trial points $w_i$ for which the empirical average $\frac{1}{m} \sum_{j=1}^{m} \ell(w_i, z_j)$ is too high.

**Lemma 4.2** (Rejection sampling). *Let $w_1, \dots, w_k$ be independent random variables satisfying $\mathbb{P}(\mathcal{L}(w_i) \leq \epsilon) \geq 1/2$. For each $i = 1, \dots, k$ draw $m$ samples $z_1, \dots, z_m \overset{i.i.d.}{\sim} \mathcal{P}$. For any $\lambda > 1$, define admissible indices $\mathcal{I} = \left\{ i \in [k] : \frac{1}{m} \sum_{j=1}^{m} \ell(w_i, z_j) \leq \lambda \epsilon \right\}$. Then with probability $1 - \exp(-\frac{k}{16}) - k \exp(-c_2 m) - \lambda^{-k/4}$, the set $\mathcal{I}$ is nonempty and $\mathcal{L}(w_i) \leq \frac{\lambda \epsilon}{c_1}$ for any $i \in \mathcal{I}$.*

We may now simply combine SGD with rejection sampling to obtain high probability guarantees. Looking at Lemma 4.2, some thought shows that the overhead for high probability guarantees is dominated by $c_2^{-1}$. As we saw from the Paley-Zygmond inequality, we always have $c_2^{-1} \lesssim D$ where $D$ upper bounds the ratios $\frac{\mathbb{E}\ell(w,z)^2}{\mathcal{L}(w)}$. It remains an interesting open question to investigate the scaling of small ball probabilities for overparametrized neural networks.

## 5 Conclusion

Existing results ensuring convergence of SGD under interpolation and the PŁ condition require the method to use a small stepsize, and therefore converge slowly. In this work we isolated conditions that enable SGD to take a large stepsize and therefore have similar iteration complexity as gradient descent. Consequently, our results align theory better with practice, where large stepsizes are routinely used. Moreover, we argued that these conditions are reasonable because they provably hold when training sufficiently wide feedforward neural networks with a linear output layer.

## 6 Acknowledgements

The work of Dmitriy Drusvyatskiy was supported by the NSF DMS 1651851 and CCF 1740551 awards. The work of Damek Davis is supported by an Alfred P. Sloan research fellowship and NSF DMS award 2047637. Yian Ma is supported by the NSF SCALE MoDL-2134209 and the CCF-2112665 (TILOS) awards, as well as the U.S. Department of Energy, Office of Science, and the Facebook Research award. Mikhail Belkin acknowledges support from National Science Foundation (NSF) and the Simons Foundation for the Collaboration on the Theoretical Foundations of Deep Learning (https://deepfoundations.ai/) through awards DMS-2031883 and #814639 and the TILOS institute (NSF CCF-2112665). This work used the programs (1) XSEDE (Extreme science and engineering discovery environment) which is supported by NSF grant numbers ACI-1548562, and (2) ACCESS (Advanced cyberinfrastructure coordination ecosystem: services & support) which is supported by NSF grants numbers #2138259, #2138286, #2138307, #2137603, and #2138296. Specifically, we used the resources from SDSC Expanse GPU compute nodes, and NCSA Delta system, via allocations TG-CIS220009.

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

# A Missing proofs

## A.1 Proof of Theorem 2.2

The proof relies on the following elementary lemma.

**Lemma A.1** (Aiming for smooth functions). *Let $\mathcal{L} \colon \mathbf{R}^d \to \mathbf{R}$ be a differentiable function. Fix two points $w, \bar{w} \in \mathbf{R}^d$ satisfying $\nabla\mathcal{L}(\bar{w}) = 0$ and $\mathcal{L}(\bar{w}) = 0$. Suppose that the Hessian $\nabla^2 \mathcal{L}$ exists and is $L$-Lipschitz continuous on the segment $[w, \bar{w}]$. Then we have*

$$\left| \mathcal{L}(w) + \frac{1}{2}\langle \nabla\mathcal{L}(w), \bar{w} - w \rangle \right| \leq \frac{5L}{12}\|w - \bar{w}\|^3.$$

*Proof.* Define the function $g(t) = \mathcal{L}(w + t(\bar{w} - w))$. The theorem is evidently equivalent to

$$\left| g(0) + \frac{1}{2}g'(0) - g(1) \right| \leq \frac{5L}{12}\|w - \bar{w}\|^3.$$

In order to establish this estimate, we first note that $g''$ is Lipschitz continuous with constant $\hat{L} := L\|w - \bar{w}\|^3$, as follows from a quick computation. Taylor's theorem with remainder applied to $g$ and $g'$, respectively, then gives

$$g(1) = g(0) + g'(0) + \frac{1}{2}g''(0) + E_1$$
$$0 = g'(1) = g'(0) + g''(0) + E_2,$$

where $|E_1| \leq \frac{\hat{L}}{6}$ and $|E_2| \leq \frac{\hat{L}}{2}$. Combining the two estimates yields

$$\begin{aligned}
g(1) &= g(0) + \frac{1}{2}g'(0) + \frac{1}{2}g'(0) + \frac{1}{2}g''(0) + E_1 \\
&= g(0) + \frac{1}{2}g'(0) - \frac{1}{2}(g''(0) + E_2) + \frac{1}{2}g''(0) + E_1 \\
&= g(0) + \frac{1}{2}g'(0) + E_1 - \frac{E_2}{2}.
\end{aligned}$$

thereby completing the proof. $\qquad\square$

Turning to the proof of Theorem 2.2, an application of Lemma A.1 guarantees

$$\langle \nabla\mathcal{L}(w), w - \bar{w} \rangle \geq 2\mathcal{L}(w) - \frac{5Lr}{6}\|w - \bar{w}\|^2,$$

for all $w \in S_r$ and any $\bar{w} \in \text{proj}_S(w)$. If the quadratic growth condition (3) is satisfied on the tube $S_r$, then we have the upper bound $\frac{5Lr}{6}\|w - \bar{w}\|^2 \leq \frac{5Lr}{3\alpha}\mathcal{L}(w)$. Theorem 2.2 follows. It remains to prove Lemma A.1.

## A.2 Proof of Lemma 2.1

In this section, we verify the classical result that the PŁ$^*$ condition on a ball implies quadratic growth. We begin with the following lemma estimating the distance of a single point to sublevel set of a function; this result is a special instance of the descent principle [6, Lemma 2.5], whose roots can be traced back to [17, Basic Lemma, Chapter 1].

**Lemma A.2** (Descent principle). *Fix a differentiable function $\mathcal{L} \colon \mathbf{R}^d \to [0, \infty)$ and a ball $B_r(w_0)$ and define $S = \{w : \mathcal{L}(w) = 0\}$. Suppose that $\mathcal{L}$ satisfies the PŁ$^*$ condition on $B_r(w_0)$ with parameter $\alpha$. Then as long as $\mathcal{L}(w_0) < \frac{1}{2}r^2\alpha$, the intersection $S \cap B_r(w_0)$ is nonempty and the estimate holds:*

$$\mathcal{L}(w_0) \geq \frac{\alpha}{2}\text{dist}^2(w_0, S).$$

*Proof.* Define the function $f(w) = \sqrt{\mathcal{L}(w)}$ and observe that for any $w \in B_r(w_0)$ with $\mathcal{L}(w) > 0$ we have $\|\nabla f(w)\|^2 = \frac{\|\nabla\mathcal{L}(w)\|^2}{4\mathcal{L}(w)} \geq \frac{\alpha}{2}$. Therefore an application of the descent principle [6, Lemma 2.5] implies that the set $S \cap B_r(w_0)$ is nonempty and the estimate $\text{dist}(w_0, S) \leq \frac{1}{\sqrt{\alpha/2}} \cdot f(w_0)$ holds. Squaring both sides completes the proof. $\qquad\square$

We may now complete the proof of Lemma 2.1 by extending from a single point $w_0$ to a neighborhood of $w_0$ as follows. First, Lemma A.2 ensures that $B_r(w_0)$ intersects $S$ at some point $\bar{w}_0$ and the inequality $\mathcal{L}(w_0) \geq \frac{\alpha}{2}\operatorname{dist}^2(w_0, S)$ holds. Fix a point $w \in B_r(w_0)$. Then clearly $\mathcal{L}$ satisfies the PŁ$^*$ condition on $B_r(w)$ with parameter $\alpha$. Let us now consider two cases: $\mathcal{L}(w) < \frac{1}{2}r^2\alpha$ and $\mathcal{L}(w) \geq \frac{1}{2}r^2\alpha$. In the former case, Lemma A.2 implies the claimed estimate $\mathcal{L}(w) \geq \frac{\alpha}{2} \cdot \operatorname{dist}^2(w, S)$. In the remaining case $\mathcal{L}(w) \geq \frac{1}{2}r^2\alpha$, we compute

$$\operatorname{dist}_S^2(w) \leq \|w - \bar{w}_0\|^2 \leq 4r^2 \leq 8\mathcal{L}(w)/\alpha.$$

Rearranging completes the proof of Lemma 2.1 .

## A.3   Proof of Lemma 2.3

Fix a point $w \in S_r$ and let $\bar{w} \in \operatorname{proj}_S(w)$ be a point satisfying the aiming condition (2.2). Observe that Lipschitz continuity of $\nabla\ell(\cdot, z)$ and interpolation ensures

$$\|\nabla\ell(w, z)\| = \|\nabla\ell(w, z) - \nabla\ell(\bar{w}, z)\| \leq \beta \cdot \operatorname{dist}(w, S) \leq \beta r.$$

Therefore for every $\tau \in [0, 1/\beta]$, the point $w^\tau := w - \tau\nabla\ell(w, z)$ satisfies

$$\operatorname{dist}(w^\tau, S) \leq \operatorname{dist}(w, S) + \tau\|\nabla\ell(w, z)\| \leq 2r.$$

Therefore the gradient $\nabla\ell(\cdot, z)$ is $\beta$-Lipschitz on the entire line segment $\{w_\tau : 0 \leq \tau \leq 1/\beta\}$. The descent lemma therefore guarantees $\|\nabla\ell(w, z)\|^2 \leq 2\beta\ell(w, z)$. Therefore upon taking expectations we obtain the second moment bound: $\mathbb{E}\|\nabla\ell(w, z)\|^2 \leq 2\beta\mathcal{L}(w)$. Next, we compute

$$\begin{aligned}
\mathbb{E}\operatorname{dist}^2(w^+, S) \leq \mathbb{E}\|w^+ - \bar{w}\|^2 &= \mathbb{E}\|(w - \bar{w}) - \eta\nabla\mathcal{L}(w, z)\|^2 \\
&= \|w - \bar{w}\|^2 - 2\eta\langle\nabla\mathcal{L}(w), w - \bar{w}\rangle + \eta^2\mathbb{E}\|\nabla\mathcal{L}(w, z)\|^2 \\
&\leq \operatorname{dist}^2(w, S) - 2\eta\theta \cdot \mathcal{L}(w) + 2\eta^2\beta\mathcal{L}(w) \quad\quad\text{(A.1)} \\
&= \operatorname{dist}^2(w, S) - 2\eta(\theta - \beta\eta)\mathcal{L}(w) \\
&\leq (1 - \alpha\eta(\theta - \eta\beta))\operatorname{dist}^2(w, S), \quad\quad\text{(A.2)}
\end{aligned}$$

where (A.1) follows from (2.2) while (A.2) follows from (2.1). The proof is complete.

## A.4   Proof of Theorem 2.4

Define the stopping time $\tau = \inf\{t \geq 1 : w_t \notin S_r\}$ and set $U_t = \operatorname{dist}^2(w_t, S)$. Note that we may equivalently write $\tau = \inf\{t \geq 1 : U_t > r^2\}$. Now, multiplying (2.3) through by $1_{\tau > t}$ yields the estimate

$$\mathbb{E}[U_{t+1}1_{\tau > t} \mid w_{1:t}] \leq (1 - \alpha\eta(\theta - \beta\eta))U_t 1_{\tau > t}. \quad\quad\text{(A.3)}$$

An application of Theorem B.2 with $q = (1 - \alpha\eta(\theta - \beta\eta))$ completes the proof.

## A.5   Proof of Theorem 2.5

We begin with the following simple lemma that bounds the second moment of the gradient estimator.

**Lemma A.3.** *For any point $w \in B_r(w_0)$, we have $\mathbb{E}\|\ell(w, z)\|^2 \leq 2\beta \cdot \mathcal{L}(w)$.*

*Proof.* Let $\bar{w}_0 \in \operatorname{proj}_S(w_0)$ be arbitrary. Observe that Lipschitz continuity of $\nabla\ell(\cdot, z)$ on $B_{3r}(w_0)$ and interpolation ensure

$$\|\nabla\ell(w, z)\| = \|\nabla\ell(w, z) - \nabla\ell(\bar{w}_0, z)\| \leq \beta \cdot \|w - \bar{w}_0\| \leq 2\beta r.$$

Therefore for every $\tau \in [0, 1/\beta]$, the point $w^\tau := w - \tau\nabla\ell(w, z)$ satisfies

$$\|w^\tau - w_0\| \leq \|w - w_0\| + \tau\|\nabla\ell(w, z)\| \leq 3r.$$

Therefore the gradient $\nabla\ell(\cdot, z)$ is $\beta$-Lipschitz on the entire line segment $\{w_\tau : 0 \leq \tau \leq 1/\beta\}$. The descent lemma therefore guarantees $\|\nabla\ell(w, z)\|^2 \leq 2\beta\ell(w, z)$. Taking the expectation of both sides completes the proof. $\qquad\square$

Next we prove the following lemma that simultaneously estimates (1) one step progress of the iterates towards $S$ and (2) how far the iterates move away from the the center $w_0$.

**Lemma A.4.** *Fix a point $w \in B_{r/3}(w_0)$ and choose $\bar{w}_0 \in \mathrm{proj}_S(w_0) \cap B_{r/3}(w_0)$. Assume $\eta \leq \frac{\theta}{\beta}$ and define $w^+ = w - \eta \nabla \ell(w, z)$ where $z \sim \mathcal{P}$. Then the following estimates hold:*

$$\mathbb{E}[\mathrm{dist}^2(w^+, S)] \leq (1 - \eta(\theta - \beta\eta)\alpha)\mathrm{dist}^2(w, S) \tag{A.4}$$

$$\mathbb{E}[\|w^+ - \bar{w}_0\|^2] \leq \|w - \bar{w}_0\|^2 + 2\eta\rho \cdot \mathrm{dist}(w, S) \tag{A.5}$$

*Proof.* Fix any point $v \in S$ and observe that

$$\|w^+ - v\|^2 = \|w - v\|^2 - 2\eta\langle\nabla\ell(w, z), w - v\rangle + \eta^2\|\nabla\ell(w, z)\|^2.$$

Taking the expectation with respect to $z$ and using Lemma A.3, we deduce

$$\mathbb{E}\|w^+ - v\|^2 \leq \|w - v\|^2 - 2\eta\langle\nabla\mathcal{L}(w), w - v\rangle + 2\beta\eta^2\mathcal{L}(w). \tag{A.6}$$

We will use this estimate multiple times for different vectors $v$.

Choose any point $\bar{w} \in \mathrm{proj}_S(w)$ satisfying the aiming condition (2.2). Define $U := \mathrm{dist}^2(w, S)$ and $U_+ := \mathrm{dist}^2(w_+, S)$. Then setting $v := \bar{w}$ in (A.6), we deduce

$$\mathbb{E}[U_+] \leq U - 2\eta\langle\nabla\mathcal{L}(w), w - \bar{w}\rangle + 2\beta\eta^2\mathcal{L}(w).$$

Using the aiming condition (2.2) we therefore deduce

$$\begin{aligned}
\mathbb{E}[U_+] &\leq U - 2\eta\theta\mathcal{L}(w) + 2\beta\eta^2\mathcal{L}(w) \\
&= U - 2\eta(\theta - \beta\eta)\mathcal{L}(w) \\
&\leq (1 - \eta(\theta - \beta\eta)\alpha)U,
\end{aligned}$$

where the last inequality follows from quadratic growth. This establishes (A.4).

Next, set $v := \bar{w}_0$ in (A.6). Defining $V = \|w - \bar{w}_0\|^2$, $V_+ = \|w^+ - \bar{w}_0\|^2$, and using Assumption 2 we therefore deduce

$$\begin{aligned}
\mathbb{E}[V_+] &\leq V - 2\eta(\theta\mathcal{L}(w) - \rho\sqrt{U}) + 2\beta\eta^2\mathcal{L}(w). \\
&= V - 2\eta(\theta - \eta\beta)\mathcal{L}(w) + 2\eta\rho\sqrt{U} \\
&= V + 2\eta\rho\sqrt{U}
\end{aligned}$$

where the last estimate follows from the inequality $\beta\eta \leq \theta$. This establishes (A.5). $\qquad\square$

For each $t \geq 0$, let $1_t$ denote the indicator of the event $E_t := \{w_0, \ldots, w_t \in B_{r/3}(w_0)\}$. Define the random variables $U_t = \mathrm{dist}^2(w_t, S)$ and $V_t = \|w_t - \bar{w}_0\|^2$. We may now multiply (A.4) by $1_{E_t}$. Noting that $1_{E_{t+1}} \leq 1_{E_t}$ we may iterate the bound yielding

$$\mathbb{E}[1_t U_t] \leq (1 - \eta(\theta - \beta\eta)\alpha)^t U_0 \tag{A.7}$$

Similarly, multiplying (A.4) by $1_{E_t}$ and using (A.7), we deduce

$$\begin{aligned}
\mathbb{E}[1_{t+1}V_{t+1}] &\leq \mathbb{E}[1_t V_t] + 2\eta\rho\mathbb{E}\sqrt{1_t U_t} \\
&\leq \mathbb{E}[1_t V_t] + 2\eta\rho(1 - \eta(\theta - \beta\eta)\alpha)^{t/2}\sqrt{U_0}
\end{aligned}$$

Iterating the recursion gives

$$\begin{aligned}
\mathbb{E}[1_t V_t] &\leq U_0 + \frac{2\rho\eta\sqrt{U_0}}{1 - \sqrt{(1 - \eta(\theta - \beta\eta)\alpha)}} \\
&\leq U_0 + \frac{4\rho\sqrt{U_0}}{(\theta - \beta\eta)\alpha}. \tag{A.8}
\end{aligned}$$

We now lower bound the probability of escaping from the ball. Note that within event $E_t$, we have

$$V_t \leq (\|w_t - w_0\| + \|w_0 - \bar{w}_0\|)^2 \leq (1 + \delta_1)^2 r^2.$$

Therefore,

$$
\begin{aligned}
\mathbb{P}(E_t^c) &\leq \mathbb{P}[V_t > (1+\delta_1)^2 r^2] \\
&\leq \mathbb{P}[V_t > (1+\delta_1)^2 r^2 \mid E_t] \cdot \mathbb{P}(E_t) \\
&\leq \frac{\mathbb{E}[1_{E_t} V_t \mid E_t]\mathbb{P}(E_t)}{(1+\delta_1)^2 r^2} \tag{A.9} \\
&\leq \frac{\mathbb{E}[1_{E_t} V_t]}{(1+\delta_1)^2 r^2} \\
&\leq \frac{U_0 + \frac{4\rho\sqrt{U_0}}{(\theta-\beta\eta)\alpha}}{(1+\delta_1)^2 r^2} \tag{A.10} \\
&\leq \left(1 + \frac{4\rho}{(\theta-\beta\eta)\alpha r}\right)\delta_1,
\end{aligned}
$$

where (A.9) follows from Markov's inequality and (A.10) follows from (A.8).

Next, we estimate the probability that $U_t$ remains small within the event $E_t$. To that end, let define the constant $C_t := (1 - \eta(\theta - \beta\eta)\alpha)^t U_0/\delta_2$. Then Markov's inequality yields

$$
\mathbb{P}(U_t > C_t \mid E_t) \leq \frac{\mathbb{E}[U_t \mid E_t]}{C_t} \leq \frac{\mathbb{E}[1_{E_t} U_t]}{C_t \mathbb{P}(E_t)} \leq \frac{\delta_2}{\mathbb{P}(E_t)}.
$$

Finally, we unconditionally bound the probability that $U_t$ remains small:

$$
\begin{aligned}
P(U_t \leq C_t) &\geq P(U_t \leq C_t \mid E_t) P(E_t) \\
&\geq \left(1 - \frac{\delta_2}{\mathbb{P}(E_t)}\right)\mathbb{P}(E_t), \\
&= \mathbb{P}(E_t) - \delta_2.
\end{aligned}
$$

as desired.

## A.6   Proof of Theorem 3.1

We first prove the following lemma, which does not require quadratic growth and relies on the Lipschitz continuity of the Jacobian $\nabla\mathcal{L}$.

**Lemma A.5.** *Consider a function $\mathcal{L}(w) = \frac{1}{2}\|F(w)\|^2$ where $F\colon \mathbf{R}^d \to \mathbf{R}^n$ is $C^1$-smooth and the Jacobian $\nabla F$ is $L$-Lipschitz on the ball $B_r(w_0)$. Then the following estimates*

$$
|\langle \nabla\mathcal{L}(w), u - v\rangle| \leq 8r^2 L\sqrt{2\mathcal{L}(w)} \tag{A.11}
$$

$$
|\langle \nabla\mathcal{L}(w), w - u\rangle - 2\mathcal{L}(w)| \leq L\sqrt{\mathcal{L}(w)/2} \cdot \|w - u\|^2, \tag{A.12}
$$

*hold for all $w, u, v \in B_r(w_0)$ satisfying $F(u) = F(v) = 0$.*

*Proof.* Fix any $w \in B_r(w_0)$ and $u, v \in B_r(w_0)$ satisfying $F(u) = F(v) = 0$. We first prove (A.11). To this end, we compute

$$
\begin{aligned}
|\langle \nabla\mathcal{L}(w), u - v\rangle| &= |\langle \nabla F(w)^\top F(w), u - v\rangle| \\
&= |\langle F(w), \nabla F(w)(u - v)\rangle| \\
&\leq \sqrt{2\mathcal{L}(w)} \cdot \|\nabla F(w)(u - v)\|,
\end{aligned}
$$

where the last estimate follows from the Cauchy-Schwarz inequality. Next, the fundamental theorem of calculus and Lipschitz continuity of $\nabla F$ yields

$$
\begin{aligned}
0 = F(v) - F(u) &= \int_0^1 \nabla F(u + t(v - u))(v - u)\, dt \\
&= \nabla F(w)(v - u) + E,
\end{aligned}
$$

where $\|E\| \leq L\|v-u\|(\|u-w\| + \|v-u\|) \leq 8r^2L$. Thus we have proved (A.11). In order to see (A.12), we compute

$$|\langle \nabla L(w), w - u\rangle - \|F(w)\|^2| = |\langle F(w), \nabla F(w)(w-u) - F(w)\rangle|$$

$$\leq \frac{L}{2}\|F(w)\|\|w-u\|^2,$$

where the last inequality follows from Cauchy-Schwarz and Lipschtiz continuity of the Jacobian $\nabla F$. Thus (A.12) holds. $\qquad\square$

Lemma A.5 quickly yields the following corollary.

**Corollary A.6.** *Consider a function* $\mathcal{L}(w) = \frac{1}{2}\|F(w)\|^2$ *where* $F\colon \mathbf{R}^d \to \mathbf{R}^n$ *is* $C^1$*-smooth and the Jacobian* $\nabla F$ *is* $L$*-Lipschitz on the ball* $B_r(w_0)$. *Suppose moreover that*

$$\mathcal{L}(w) \geq \frac{\alpha}{2} \cdot \mathrm{dist}^2(w, S)$$

*where* $S = \{w : F(w) = 0\}$. *Then the estimates*

$$\langle \nabla \mathcal{L}(w), w - \bar{w}\rangle \geq 2\mathcal{L}(w) - \frac{L\sqrt{2}}{\alpha} \cdot \mathcal{L}(w)^{3/2}. \tag{A.13}$$

$$\langle \nabla \mathcal{L}(w), w - v\rangle \geq 2\mathcal{L}(w) - \frac{L\sqrt{2}}{\alpha} \cdot \mathcal{L}(w)^{3/2} - 8r^2L\sqrt{2\mathcal{L}(w)}, \tag{A.14}$$

*hold for all* $w \in B_r(w_0)$, $v \in B_r(w_0) \cap S$, *and* $\bar{w} \in B_r(w_0) \cap \mathrm{proj}_S(w)$.

*Proof.* We will apply Lemma A.5. Setting $u := \bar{w}$ in Lemma A.12 and using quadratic growth yields the estimate:

$$\langle \nabla \mathcal{L}(w), w - \bar{w}\rangle \geq 2\mathcal{L}(w) - L\sqrt{\mathcal{L}(w)/2} \cdot \|w - \bar{w}\|^2$$

$$\geq 2\mathcal{L}(w) - \frac{L}{\alpha}\sqrt{2\mathcal{L}(w)} \cdot \mathcal{L}(w),$$

which establishes (A.13). Adding this estimate to (A.12) with $u = \bar{w}$ yields (A.14). $\qquad\square$

Theorem 3.1 follows immediately from the corollary. Indeed, note that let $\bar{w} \in \mathrm{proj}_S(w)$ satisfies $\|w - \bar{w}\| \leq 2r$. Therefore we may apply Corollary A.6 and use the estimate $\mathcal{L}(w) \leq \frac{\beta}{2}\|w - \bar{w}\|^2$.

### A.7 Proof of Theorem 3.3

Theorem 3.2 implies that with probability at least $1 - p - 2\exp(-\frac{ml}{2}) - (1/m)^{\Theta(\ln m)}$, the Jacobian of $\nabla F$ is Lipschitz with constant $L = \tilde{O}\left(\frac{r^{3l}}{\sqrt{m}}\right)$ on $B_r(w_0)$ and the PŁ-condition holds on $B_{2r}(w_0)$ with parameter $\lambda_0/2$. Using the assumption $r = \Omega(\frac{1}{\sqrt{\lambda_0}})$, we may apply Lemma 2.1 to deduce quadratic growth. Next, we will apply Theorem 3.1 in order to deduce (uniform) aiming with parameters $\theta = 1$. To this end, it suffices to ensure $2rL\sqrt{\beta}/\lambda_0 \leq 1$, which follows from the prerequisite assumption $m = \Omega(\frac{r^{6l+2}}{\alpha^2})$.

To see the last claim, for any $w \in B_r(w_0)$ we compute the Hessian

$$\nabla^2 \ell_i(w) = \nabla f(w, x_i)\nabla f(w, x_i)^T + (f(w, x_i) - y_i)\nabla^2 f(w, x_i).$$

Therefore,

$$\|\nabla^2 \ell_i(w)\|_{\mathrm{op}} \leq \|\nabla f(w, x_i)\|^2 + |f(w, x_i) - y_i| \cdot \|\nabla^2 f(w, x_i)\|_{\mathrm{op}}. \tag{A.15}$$

Setting $\bar{w}_0 \in \mathrm{proj}_S(w_0) \cap B_r(w_0)$, observe that

$$\|\nabla f(w, x_i)\| = \|\nabla f(w, x_i) - \nabla f(\bar{w}_0, x_i)\| \leq 2r \sup_{v \in B_r(w_0)} \|\nabla^2 f(v, x_i)\|_{\mathrm{op}} = \tilde{O}\left(\frac{r^{3L+1}}{\sqrt{m}}\right),$$

where the last inequality follows from transition to linearity (3.1). Thus $f(\cdot, x_i)$ is Lipschitz continuous on the ball $B_r(w_0)$. Therefore, we may also bound

$$|f(w, x_i) - y_i| = |f(w, x_i) - f(\bar{w}, x_i)| = \tilde{O}\left(\frac{r^{3L+2}}{\sqrt{m}}\right)$$

Returning to (A.15), we thus have $\|\nabla^2 \ell_i(w)\|_{\mathrm{op}} = \tilde{O}\left(\frac{r^{6L+2}}{m}\right)$. Taking into account that $m = \tilde{\Omega}\left(\frac{nr^{6l+2}}{\lambda_0^2}\right)$ we deduce $\frac{r^{6L+2}}{m} = O(1)$ thereby completing the proof.

## A.8 Proof of Corollary 3.4

The goal is to apply Theorem 2.5. To this end, we begin by applying Theorem 3.3. We may then be sure that with high probability Assumptions 1 and 2 hold on a ball $B_r(w_0)$ with $\alpha = \lambda_0/2$, $\theta = 1$, and $\rho = \tilde{O}\left(\frac{r^{3l+2}}{\sqrt{m}}\right)$. We may then choose $\eta = \frac{1}{2\beta} = \Theta(1)$. In order to make sure that $\rho \leq (\theta - \beta\eta)\alpha r$, it suffices to be in the regime $m = \Omega(\frac{r^{6l+2}}{\lambda_0^2})$. Finally it remains to ensure that $\mathrm{dist}^2(w_0, S) \leq \delta_1^2 r^2$. To do so, using quadratic growth, we have $\mathrm{dist}^2(w_0, S) \leq \frac{4}{\lambda_0}\mathcal{L}(w_0) = O(\frac{1}{\lambda_0})$. Thus it suffices to let $r = \Omega(\frac{1}{\delta_1\sqrt{\lambda_0}})$. An application of Theorem 2.5 completes the proof.

## A.9 Proof of Lemma 4.2

Define the set $\mathcal{J} = \{i \in [k] : f(w_i) \leq \epsilon\}$. Then Hoeffding inequality ensures that the inequality $|\mathcal{J}| \geq \frac{k}{4}$ holds with probability at least $1 - \exp(-k/16)$. Conditioned on this event, consider any index $i \in [k]$ satisfying $\mathcal{L}(w_i) > \frac{\lambda\epsilon}{c_1}$. Then from (3), we know that with probability at least $1 - \exp(-c_2 m)$, we have

$$\frac{1}{m}\sum_{i=1}^{m} \ell(w_i, z_i) \geq c_1\mathcal{L}(w_i) > \lambda\epsilon,$$

and therefore $i \notin \mathcal{I}$. Let us now estimate the probability that $\mathcal{I}$ is nonempty conditioned on $|\mathcal{J}| \geq \frac{k}{4}$. To this end, by Markov's inequality for any $i \in \mathcal{J}$, we have

$$P\left(\frac{1}{m}\sum_{j=1}^{m}\ell(w, z_j) > \lambda\epsilon\right) \leq P\left(\frac{1}{m}\sum_{j=1}^{m}\ell(w_i, z_j) \geq \lambda\mathcal{L}(w_i)\right) \leq \frac{\mathcal{L}(w_i)}{\lambda f(w_i)} = \frac{1}{\lambda}.$$

Consequently, the probability that the set $\mathcal{I}$ is empty is at most $\lambda^{-k/4}$.

## B Auxiliary results on stopping times

**Theorem B.1** (Stopping time argument). *Let $\{U_t\}_{t\geq 0}$ be a sequence of nonnegative random variables and define the stopping time $\tau = \inf\{t \geq 0 : U_t > u\}$ for some constant $u > 0$. Suppose that*

$$\mathbb{E}[U_{t+1}1_{\tau > t} \mid U_{1:t}] \leq U_t 1_{\tau > t} + \zeta_t \qquad \forall t \geq 0. \tag{B.1}$$

*Then the estimate $\mathbb{P}[\tau \leq t] \leq \frac{\mathbb{E}U_0 + \sum_{i=0}^{t-1}\zeta_i}{u}$ holds for all $t \geq 1$.*

*Proof.* Observe that by Markov's inequality, we have

$$\mathbb{P}[\tau \leq t] = \mathbb{P}[U_{t\wedge\tau} > u] \leq \frac{\mathbb{E}[U_{t\wedge\tau}]}{u}.$$

Let us therefore bound the expectation of the stopped random variable $V_t := U_{t\wedge\tau}$. Letting $\mathbb{E}_t[\cdot]$ denote the conditional expectation $\mathbb{E}[\cdot \mid U_{1:t}]$, we successively compute

$$\begin{aligned}
\mathbb{E}_t[V_{t+1}] &= \mathbb{E}_t[U_{t+1\wedge\tau}] \\
&= \mathbb{E}_t[U_{t+1\wedge\tau}1_{\tau>t}] + \mathbb{E}_t[U_{t+1\wedge\tau}1_{\tau\leq t}] \\
&= \mathbb{E}_t[U_{t+1}1_{\tau>t}] + \mathbb{E}_t[U_{t\wedge\tau}1_{\tau\leq t}] \\
&= \mathbb{E}_t[U_{t+1}1_{\tau>t}] + U_{t\wedge\tau}1_{\tau\leq t} \\
&\leq U_t 1_{\tau>t} + U_{t\wedge\tau}1_{\tau\leq t} + \zeta_t \\
&= V_t + \zeta_t.
\end{aligned}$$

Taking the expectation with respect to $U_{1:t}$, applying the tower rule, and iterating the recursion we deduce $\mathbb{E}[V_t] \leq \mathbb{E}[U_0] + \sum_{i=0}^{t-1}\zeta_i$, thereby completing the proof. $\qquad\square$

**Theorem B.2** (Stopping time with contractions). *Let $\{U_t\}_{t\geq 0}$ be a sequence of nonnegative random variables and define the stopping time $\tau = \inf\{t \geq 0 : U_t > u\}$ for some constant $u > 0$. Suppose that there exists $q \in (0,1)$ such that*

$$\mathbb{E}[U_{t+1}1_{\tau>t} \mid U_{1:t}] \leq q \cdot U_t 1_{\tau>t} \qquad \forall t \geq 0. \tag{B.2}$$

*Then as long as $\mathbb{E}U_0 \leq \delta_1 u$, the event $\{\tau = \infty\}$ occurs with probability at least $1 - \delta_1$. Moreover, with probability at least $1 - \delta_1 - \delta_2$, the estimate $U_t \leq \varepsilon U_0$ holds after $t \geq \frac{1}{1-q}\log\left(\frac{1}{\delta_2 \varepsilon}\right)$ iterations.*

*Proof.* Define the stopping time $\tau = \inf\{t \geq 1 : U_t > u\}$. An application of Lemma B.1 therefore implies $\mathbb{P}[\tau < t] \leq \frac{\mathbb{E}U_0}{u} \leq \delta_1$. Taking the limit as $t \to \infty$, we deduce that the event $\{\tau = \infty\}$ occurs with probability at least $1 - \delta_1$. Next taking the expectation with respect to $U_{1:t}$ in (B.2) and applying the tower rule gives

$$\mathbb{E}[U_{t+1}1_{\tau > t}] \leq q \cdot \mathbb{E}[U_t 1_{\tau > t}].$$

Taking into account that $1_{\tau > t} \geq 1_{\tau > t+1}$ we may iterate the recursion thereby yielding

$$\mathbb{E}[U_t 1_{\tau > t}] \leq q^t U_0.$$

Now, setting $\varepsilon' := \varepsilon U_0$, Markov's inequality yields

$$\mathbb{P}[U_t 1_{\tau > t} \geq \varepsilon'] \leq \frac{\mathbb{E}[U_t 1_{\tau > t}]}{\varepsilon'} \leq \frac{q^t U_0}{\varepsilon'} \leq \delta_2.$$

Finally observe

$$\mathbb{P}[U_t \geq \varepsilon' \mid \tau = \infty] = \frac{\mathbb{P}(U_t \geq \varepsilon', \tau = \infty)}{\mathbb{P}[\tau = \infty]} \leq \frac{\mathbb{P}(U_t 1_{\tau > t} \geq \varepsilon')}{\mathbb{P}[\tau = \infty]}.$$

Therefore we deduce

$$\begin{aligned}
\mathbb{P}[U_t < \varepsilon'] &\geq \mathbb{P}[U_t < \varepsilon' \mid \tau = \infty] \cdot \mathbb{P}[\tau = \infty] \\
&= (1 - \mathbb{P}[U_t \geq \varepsilon' \mid \tau = \infty]) \cdot \mathbb{P}[\tau = \infty] \\
&\geq \mathbb{P}[\tau = \infty] - \mathbb{P}[U_t \geq \varepsilon' \mid \tau = \infty] \cdot \mathbb{P}[\tau = \infty] \\
&\geq \mathbb{P}[\tau = \infty] - \mathbb{P}(U_t 1_{\tau > t} \geq \varepsilon') \\
&\geq 1 - \delta_1 - \delta_2,
\end{aligned}$$

as claimed. $\square$

## C  A bad example

**Lemma C.1.** *Consider the objective $\mathcal{L}(x,y) = \frac{1}{2}(y - ax^2)^2$ for any $a > 0$. Then $\mathcal{L}$ is $C^\infty$ near $(0,0)$ and the function $\mathcal{L}$ satisfies the PŁ inequality (PŁ) with constant $\alpha = 1$. Therefore $\mathcal{L}$ satisfies the aiming condition on some neighborhood of the origin. However, for any neighborhood $U$ of the origin, the function $\mathcal{L}$ is not quasar-convex on $U$ relative to any point $(x, ax^2) \in U$.*

*Proof.* To see the validity of the PŁ-condition, observe that

$$\frac{\|\nabla \mathcal{L}(x,y)\|}{2\sqrt{\mathcal{L}(x,y)}} = \|\nabla \sqrt{\mathcal{L}(x,y)}\| = \tfrac{1}{\sqrt{2}}\|\nabla |y - ax^2|\| \geq \tfrac{1}{\sqrt{2}}.$$

It follows immediately from Lemma 2.2 that $\mathcal{L}$ satisfies the aiming condition (2.2) on some neighborhood of the origin. Next we verify the failure of quasar convexity. Consider an arbitrary point $(x, ax^2)$ on the parabola. For any $(u, v) \in \mathbf{R}^2$, we compute

$$\begin{aligned}
\langle \nabla \mathcal{L}(u,v), (u,v) - (x, ax^2)\rangle &= \langle (-2au, 1), (u - x, v - ax^2)\rangle \cdot (v - au^2) \\
&= (-2au(u - x) + v - ax^2)(v - au^2).
\end{aligned}$$

In particular, setting $z_\gamma = (0, \gamma)$ we obtain

$$\langle \nabla \mathcal{L}(z_\gamma), z_\gamma - (x, ax^2)\rangle = \gamma^2 - \gamma ax^2.$$

Note the right hand side is negative for any $0 < \gamma < ax^2$. Thus, letting $\gamma > 0$ tend to zero, we deduce that $\mathcal{L}$ is not quasar-convex relative to $(x, ax^2) \in U$ with $x \neq 0$. Let us consider now the setting $x = 0$. Then we compute

$$\langle \nabla \mathcal{L}(u,v), (u,v) - (0,0)\rangle = (v - 2au^2)(v - au^2).$$

The right side is negative if $au^2 < v < 2au^2$ and therefore $\nabla \mathcal{L}$ is not quasar-convex relative to $(0,0)$ on any neighborhood of $(0,0)$. $\square$

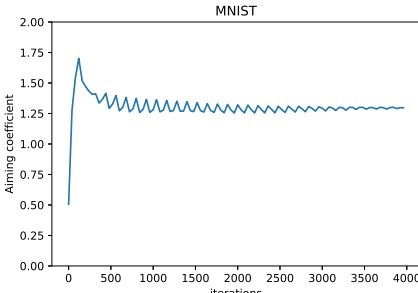
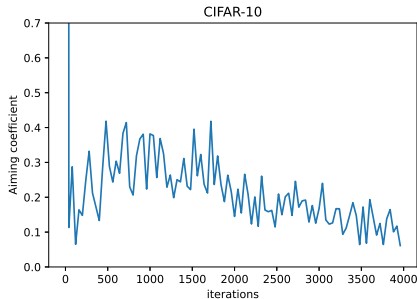

Figure 4: Verifying aiming condition. Left: MNIST on fully-connected neural network; Right: CIFAR-10 on CNN.

## D  Numerical verification of aiming condition on neural networks

In principle, the aiming condition is difficult to verify, as it involves $proj_S(w)$ – the nearest point of $w$ to the solution set $S$, which we have no way of computing. Instead, we replace $proj_S(w)$ by the last iterate $\bar{w}$ of an SGD run, and compute an estimate of the aiming condition coefficient by the quotients $\hat{\theta}_t \triangleq \langle \nabla L(w_t), w_t - \bar{w} \rangle / L(w_t)$ along the iterate path. Note that this quotient is not the same quotient as would appear in the definition of quasar-convexity because $\bar{w}$ is a random point that depends on the iterate path taken by SGD.

*Settings:* We conduct the experiments on two datasets, MNIST and CIFAR-10. For MNIST, we trained a 3-hidden layer fully-connected neural network (width$= 1000$). First, we train the network until convergence (loss function value smaller than $10^{-4}$), and record the parameters of this trained network as the optimal solution $\bar{w}$. For CIFAR-10, we train a two-layer CNN for $4000$ iterations, and record the final parameter setting as $\bar{w}$. Then, for each experiment, we took a second run, and at each iteration $w_t$ we compute the estimate of the aiming condition coefficient using the equation mentioned above.

*Results:* Figure 4 show the plots of against the iteration number. In both cases, we observe that the estimate stays positive, which suggests that aiming condition holds.

## E  Aiming Condition vs. quasar-convexity

Following [15], a nonnegative function $\mathcal{L}$ is called *quasar-convex relative to a point* $\bar{w}$ with $\mathcal{L}(\bar{w}) = 0$ if the inequality $\langle \nabla \mathcal{L}(w), w - \bar{w} \rangle \geq \theta \cdot \mathcal{L}(w)$ holds for all $w$ near $\bar{w}$. Aiming, in contrast, stipulates the analogous inequality but with $\bar{w}$ crucially replaced by $proj_S(w)$, where $S$ is the set of zero loss solutions. We argue that in the interpolation setting, quasar-convexity is a very restrictive condition because it requires the set of zero-loss solutions to be an affine subspace, which is usually not the case (see for example [24]). To see this, note that quasar-convexity implies that $S$ is star-convex relative to $\bar{w}$ (Observation 3 in [15]). That is, there exists $\epsilon > 0$ such that for any $w \in B_\epsilon(\bar{w}) \cap S$, the line segment joining $w$ and $\bar{w}$ is fully contained in $S$. This is a very strong conclusion. For example, a smooth-manifold $S$ is star convex near $\bar{w}$ if and only if it coincides with a linear subspace around $\bar{w}$. The PL-condition in turn implies by Theorem 2.16 in [35] that $S$ is a smooth manifold. Therefore, if loss functions for training wide neural networks were quasar-convex, then the set of interpolating solutions would form a linear space on any compact set, which is certainly not true. In contrast, as we have proved, any $C^3$-smooth function satisfying PL automatically satisfies the aiming inequality near a point $\bar{w} \in S$. In summary, quasar-convexity does not hold when training wide neural networks (since otherwise it would imply that interpolating solutions form a linear subspace), while aiming provably holds—one of our main results.

