# OpenReview forum: "Aiming towards the minimizers: fast convergence of SGD for overparametrized problems"
_NeurIPS.cc/2023/Conference — NeurIPS 2023 poster_

### Official Review · Reviewer_APCi · 2023-07-06

**Soundness:** 3 good
**Presentation:** 3 good
**Contribution:** 3 good
**Rating:** 6
**Confidence:** 3

**Summary:**

This work proposes a new condition called the aiming condition, which looks similar to quasar-convexity but provides fundamentally different convergence guarantees for SGD. Under the aiming condition, along with several other regularity conditions, SGD can achieve the same sample complexity as GD. It is then shown that wide neural networks enjoys this property with high probability.

**Strengths:**

- The aiming condition is impressive as a condition for SGD to achieve the same sample complexity as GD.
- The presentation of the results is clear. This work embodies a lot of results, which could potentially make the paper hard to follow. However, the introduction part provides a clear roadmap.

**Weaknesses:**

- I am expecting more discussions on the comparison of the aiming condition against existing conditions (e.g. quasar-convexity).This involves two aspects: 1. I would like to see some more examples where the aiming condition is satisfies but quasar-convexity does not hold; 2. It would be interesting to provide some intuition about why quasar-convexity cannot provide a similar result.
- This work does not seem to be enough strong technically.
- This work has some minor problems presentations. For example, the same sentence appears twice in Line 62 and Line 160; Furthermore, in Line 251, the radius needs to be scaled up rather than shrunk.

**Questions:**

I am mainly curious about the comparison of quasar-convexity and the aiming condition (see "Weaknesses" above).

**Limitations:**

This work, with a theorical nature, does not have potential negative societal impact.

---

> ### Author Rebuttal · Authors · 2023-08-08
>
> We thank the reviewer for the feeback. We address your concerns below.
>
> **W1:** *I am expecting more discussions on the comparison of the aiming condition against existing conditions (e.g. quasar-convexity).This involves two aspects: 1. I would like to see some more examples where the aiming condition is satisfies but quasar-convexity does not hold; 2. It would be interesting to provide some intuition about why quasar-convexity cannot provide a similar result.*
>
> **A:** Thank you for bringing attention to this point.  We will now argue  that in the interpolation setting, quasar-convexity is a very restrictive condition because it requires the set of zero-loss solutions to be a linear subspace, which is usually not the case (see for example [1]).
> To be precise, quasar-convexity relative to a point $\bar w$ with $L(\bar w)=0$ means that $\langle \nabla L(w),w-\bar w\rangle\geq \theta\cdot L(w)$ for all $w$ near $\bar w$. Aiming, in contrast, stipulates the analogous inequality but with $\bar w$ crucially replaced by $proj_S(w)$, where $S$ is the set of zero loss solutions.
> To see the limitation of quasar-convexity, note that quasar-convexity implies that $S$ is star-convex relative to $\bar w$ (Observation 3 in reference [13] in the paper).
>  That is, there exists $\epsilon>0$ such that for any $w\in B_\epsilon(\bar w)\cap S$, the line segment joining $w$ and $\bar w$ is fully contained in $S$. This is a very restrictive assumption. For example, a smooth-manifold $S$ is star convex near $\bar w$ if and only if it coincides with a linear subspace around $\bar w$. The PL-condition in turn implies by Theorem 2.16 in [2] that $S$ is a smooth manifold. Therefore, if loss functions for training wide neural networks were  quasar-convex, then the set of interpolating solutions would form a linear space on any compact set, which is certainly not true. In contrast, as we have proved, on any fixed radius ball around the initial point, sufficiently wide neural networks satisfy the aiming condition. In parallel, we also showed that any $C^3$-smooth function satisfying PL automatically satisfies the aiming inequality near a point $\bar w\in S$. In summary, quasar-convexity does not hold when training wide neural networks (since otherwise it would imply that interpolating solutions form a linear subspace), while aiming provably holds---one of our main results. We will add a discussion along these lines to the revision.
>
>
> [1] Liu, Zhu, Belkin, Loss landscapes and optimization in over-parameterized non-linear systems and neural networks, ACHA, 2022.
>
> [2] Rebjock, Boumal, Fast convergence to non-isolated minima:
> four equivalent conditions for $C^2$ functions, arxiv.org/pdf/2303.00096.pdf.
>
> **W2:** *This work does not seem to be enough strong technically.*
>
> **A:**  In this paper, we address a fundamental question of interest both for optimization and machine learning: why is that numerically the iteration complexity of SGD is comparable to GD when training wide neural networks. In order to answer this question, we introduced a number of new analysis techniques. First, the proof that neural networks satisfy aiming and uniform aiming---new conditions introduced in the paper---relied on some careful computations with nonlinear least squares and the transition to linearity phenomenon. Second, the proof of convergence of SGD relied on decomposing the iterate trajectories into "tangent" and "orthogonal" components to the solution set. The aiming condition ensured that the iterates make steady progress in orthogonal directions towards while uniform aiming controlled the tangent motion. Third, using a novel martingale stopping time argument together with this decomposition, we bounded the probability that the iterates escape the ball. We expect that such an orthogonal decomposition together with a stopping time argument to be useful more broadly.
>
> With these new techniques, we were able to contribute the following novelties. We proved that sufficiently wide neural networks satisfy our proposed aiming condition. Note that the width requirement we have is the same as that in establishing convergence of GD in prior work; hence, we are not imposing stronger assumptions. Second, we proved that aiming, interpolation, and quadratic growth conditions enable application of SGD with a large step size and fast linear rate. Indeed, Section 1.2 explains in detail that the linear rate we obtain is much faster than existing convergence guarantees for SGD. We believe that this paper is strong technically, and the developed techniques and regularity conditions will be useful for understanding optimization algorithms and landscapes of loss functions in nonconvex optimization and learning.
>
> **W3:** *The same sentence appears twice in Line 62 and Line 160; Furthermore, in Line 251, the radius needs to be scaled up rather than shrunk.*
>
> **A:** Thank you for pointing this out. We will correct them in the revision and do a thorough pass through the paper.

---

> > ### Comment · Reviewer_APCi · 2023-08-17
> > **Thanks for the rebuttal**
> >
> > I would like to thank the authors for the detailed rebuttal. I certainly had some misunderstandings when writing the initial review, and I appreciate the authors for the great explanation. I will vote for this work being accepted.

---

### Official Review · Reviewer_vP9H · 2023-07-06

**Soundness:** 4 excellent
**Presentation:** 3 good
**Contribution:** 3 good
**Rating:** 6
**Confidence:** 3

**Summary:**

This work shows a regularization condition for SGD in the interpolation regime which allows it to have same fast linear convergence rate as deterministic gradient descent. Hence, the theory presented in this paper supports the practical observation that with the same (large) learning rate, mini-batch SGD has almost the same convergence rate of SGD. This goes against the traditional approaches for SGD convergence under PL condition that require SGD to have smaller step-size than GD, hence making SGD's convergence slower in theory. The authors here present the conditions in which SGD have similar iteration complexity as GD.

**Strengths:**

1) The main strength of the work lies in it's novelty. The motivation of the problem seems clear from the introduction.
2) This is an important problem since it is crucial to have a large learning rate for SGD as it helps improving generalization. Hence, the current theories that analyze SGD under PL-condition and requires it to have a smaller l.r, may suffer from having good generalization. The theory presented by the authors allows the same large learning rate to be used as GD. Hence, the variance of SGD coupled with the alowable large learning rate can boost generalization.

**Weaknesses:**

1) Looking at the theorems, I still feel the assumptions made are too strong too hold in a non-convex landscape especially Aiming. For theorem-1.2, which consdiers a non-convex loss landscape on $w$, it is unclear how aiming can hold on a ball of radius $r$ and it's relation to the curvature at that point (given by minimal eigenvalue of NTK). The authors make sure that the iterates remain inside a ball $B_{r}(w_{0})$ near the minima but the radius depends on the eigenvalue of NTK at initializaiton! This makes a very strong assumption that the initial point is set to be very close to the true minima $w_{0}$. More so, if initialization is made far away from the minima, there is a high probability that aiming does not hold.

2) From the theorems, it is unclear that how the stochasticity from the mini-batch gradients effect the convergence or what is it's effect in ensuring that the iterates don't escpae $B_{r}(w_{0})$. For theorem-2.3, I assume that it will take longer iterations if the variance in mini-batch gradient is high. It is not very clear from the inequality as how the relation is.

A minor correction:

3) On line-42, it might be better to cite a different source such as [1] to refer to the generalization effect of large learning rate, as the authors of "Edge of stability" themselves mention that they don't claim the generalization effect of large learning rate but rather focus on the stability effect. See https://youtu.be/6xeh6gfESuc?t=3016

[1] Li, Yuanzhi, Colin Wei, and Tengyu Ma. "Towards explaining the regularization effect of initial large learning rate in training neural networks." Advances in Neural Information Processing Systems 32 (2019).

**Questions:**

See points 1 and 2.

**Limitations:**

The main limitation of this work is a strong assumption on the locality of the SGD analysis. Some further clarification would be helpful.

---

> ### Author Rebuttal · Authors · 2023-08-08
>
> We thank the reviewer for the positive feedback. We address the concerns below:
>
> **W1:** *Looking at the theorems, I still feel the assumptions made are too strong too hold in a non-convex landscape especially Aiming. For theorem-1.2, which consdiers a non-convex loss landscape on $w$, it is unclear how aiming can hold on a ball of radius $r$
>  and it's relation to the curvature at that point (given by minimal eigenvalue of NTK). The authors make sure that the iterates remain inside a ball $B_r(w_0)$ near the minima but the radius depends on the eigenvalue of NTK at initializaiton! This makes a very strong assumption that the initial point is set to be very close to the true minima $w_0$. More so, if initialization is made far away from the minima, there is a high probability that aiming does not hold.*
>
> **A:** Thank you for bringing this to our attention; we can see where the confusion arises in the informal statement of Theorem 1.2.
>  In the statement of the theorem, one can replace $\lambda$---the minimal eigenvalue of the NTK at initialization---by the the minimal eigenvalue $\lambda_{\infty}$ of the NTK of an infinitely wide neural network, which *does not* depend on the initialization. See the discussion on page 7 for a precise statement. The two theorem statements are equivalent  because as the width increases, the parameter $\lambda$ approaches $\lambda_{\infty}$. More precisely, in the regime $m=\Omega(nr^{6l+2}/\lambda_{\infty}^2)$, with high probability one has $\lambda\geq \lambda_{\infty}/2$. Therefore the requirement on the radius $r= \Omega(1/\lambda \delta^2)$ is independent of the initialization, with high probability. Summarizing,  *for any fixed radius $r= \Omega(1/\lambda_{\infty} \delta^2)$, one may take the width sufficiently large $m=\Omega(nr^{6l+2}/\lambda_{\infty}^2)$, so that with probability roughly $1-\delta$ aiming holds on the entire ball $B_r(w_0)$ and the SGD iterates converge at the linear rate $O(\exp(-t\lambda_{\infty}))$.*
>
> We emphasize that in this parameter regime (i.e. when $m$ is sufficiently large), the ball $B_r(w_0)$ is in fact large enough to contain the whole optimization path of gradient descent (see for example Section 5 in [1]), and the initial point need *not* be close to the true minima. That means, even if "initialization is made far away from the minima", $B_r(w_0)$ still covers both the initialization and the global minima. Hence, the aiming condition would hold throughout the training process.
>
> We will modify the statement of the theorem and the preceding discussion to make these points clear. We hope that this explanation convinces the reviewer that aiming is indeed a reasonable condition.
>
> [1] Liu, Zhu, Belkin, Loss landscapes and optimization in over-parameterized non-linear systems and neural networks, ACHA, 2022.
>
> **W2:** *From the theorems, it is unclear that how the stochasticity from the mini-batch gradients effect the convergence or what is it's effect in ensuring that the iterates don't escape $B_r(w_0)$. For theorem-2.3, I assume that it will take longer iterations if the variance in mini-batch gradient is high. It is not very clear from the inequality as how the relation is.*
>
> **A:** Thank you for the question. Interpolation implies the basic bound $E[||\nabla \ell(w,z)||^2] \le 2\beta \cdot L(w)$, where $\beta$ is the smoothness constant of the loss functions (Line 408 of our appendix). Thus the second moment of the stochastic gradient (and its variance) tends to zero as the iterates approach the solution set $S$. The "rate" at which the variance tends to zero is controlled by $\beta$. Now, if one wants to use this bound in order to ensure that the expected function value $E[L(w_t)]$ contracts in each iteration, one would need to use a tiny learning rate $\eta=\frac{\alpha}{\beta^2}$ leading to a slow rate of convergence (as simple examples show).  To overcome this difficulty, we focus on the contraction in the squared distance $E[dist^2(w_t,S)]$. The aiming condition together with the aforementioned bound on the variance ensures that $E[dist^2(w_t,S)]$ indeed contracts in each iteration when using a stepsize on the order $\eta=\frac{\theta}{\beta}$ and while the iterates remain in $B_r(w)$. That is, the the iterates makes very good progress moving "orthogonally" towards $S$.
>  It remains to bound the probability that the iterates escape the ball $B_r(w)$. This bit of the argument is involved. We first establish an auxiliary condition called uniform aiming (page 6) for wide NNs, and use it to bound the "tangent motion" relative to $S$. A careful stopping time argument then bounds the probability of escaping the ball. We will add a more detailed discussion to the revision.
>
> **W3:** *On line-42, it might be better to cite a different source such as [1] to refer to the generalization effect of large learning rate$\ldots$*
>
> **A:** Thank you for pointing this out. We will cite this reference instead.

---

> > ### Comment · Reviewer_vP9H · 2023-08-19
> >
> > i acknowledge that I have read the author's response and they have answered my concerns. I intend to keep my score.

---

### Official Review · Reviewer_FvUq · 2023-07-07

**Soundness:** 3 good
**Presentation:** 3 good
**Contribution:** 3 good
**Rating:** 6
**Confidence:** 3

**Summary:**

This work presents a set of conditions under which the convergence rate of SGD with large step size is similar to that of gradient descent (the deterministic setting). This is in contrast to prior work, where the convergence of over-parameterized SGD under PL condition require small step size, and converge slowly as a result of this small step size.



**Strengths:**

1. The paper is well-motivated and clearly organized. This work provides a theoretical result that improve previous analyses of the linear convergence rate of overparameterized SGD.
2. The paper thoroughly compares its result to prior works mentioned.
3. The paper attempts verifies assumptions that it makes.


**Weaknesses:**

1. The paper states that its goal is to improve stepsize selection and convergence rate of SGD for nonconvex problems under certain conditions (see lines 133-134). However, the paper does not directly do this.
2. This work claims to be the first to present an analysis of the convergence rate of over-parameterized SGD with large step size which is similar the convergence rate attained by gradient descent,. However, prior work touches upon this as well (see “Observations in simplified settings” in The Impact of Neural Network Overparameterization on Gradient Confusion and Stochastic Gradient Descent by Sankararaman, De, Xu, Huang, and Goldstein). Perhaps it is worth connecting both works to further demonstrate this paper’s contribution. (Please correct me if I am mistaken.)
3. While the contribution of this paper is predominantly theoretical, I believe that this work could benefit from some empirical evaluation as well. Current experiments are fairly limited (MNIST is the only dataset used and is easy to learn. Additionally, model architecture is also quite limited. Using other datasets e.g. CIFAR10/100 or running the experiments for networks with varying widths might be useful).


**Questions:**

There is no figure 2


**Limitations:**

The authors have described the limitations.

---

> ### Author Rebuttal · Authors · 2023-08-08
>
> We thank the reviewer for the positive comments. We address your concerns below in detail.
>
> **W1:** *The paper states that its goal is to improve stepsize selection and convergence rate of SGD for nonconvex problems under certain conditions (see lines 133-134). However, the paper does not directly do this.*
>
> **A:** Apologies; this statement was not carefully stated. In line 133-134, we mean to state our goal of obtaining *theoretically acceptable* stepsizes, which better match the large stepsizes used in practice.
> In this paper, we successfully showed that, under certain conditions (including aiming and PL), our theory indeed allows larger stepsizes than prior works and leads to a significantly faster convergence rate, as explained in Section 1.2. Moreover, we proved that these conditions hold for sufficiently wide neural networks.
>
> **W2:**  *This work claims to be the first to present an analysis of the convergence rate of over-parameterized SGD with large step size which is similar the convergence rate attained by gradient descent. However, prior work touches upon this as well (see “Observations in simplified settings” in The Impact of Neural Network Overparameterization on Gradient Confusion and Stochastic Gradient Descent by Sankararaman, De, Xu, Huang, and Goldstein). Perhaps it is worth connecting both works to further demonstrate this paper’s contribution. (Please correct me if I am mistaken.)*
>
> **A:** Thank you for pointing out this reference (We call it "the reference" later on). We agree that this reference is relevant and indeed has some discussion on large step size for SGD. We will be sure to cite it in our revision and include a comparison.
>
> We summarize the differences as follows:
>
> 1. (PL condition for each summand) The reference requires that every loss function $f_i$, $i\in[N]$, satisfies the $\mu$-PL condition (see (A2) in page 3 of the reference), which is {\em much stronger} than the usual PL condition for the ERM loss $F=\frac{1}{N}\sum_{i=1}^N f_i$, which we use in our paper. Indeed, it is not clear what the limiting value of $\mu$ in the reference would be as the width increases to infinity. In contrast, the PL constant that appears in our bounds is the minimal eigenvalue of the NTK of the infinitely wide neural network.
>
>
> 2. (slower convergence rate) According to Theorem 3.1 in the reference,  the required stepsize is $\alpha=1/N L$, which leads to a slow rate of convergence $O(\frac{N^2L}{\mu}\log(1/\epsilon))$ to a fixed constant loss value $\epsilon\approx \eta N/\mu$, which itself can be quite large. Here, $\eta$ is a gradient confusion constant. In contrast, the number of samples $N$ does not influence the convergence rate we obtain, and we prove fast convergence to the global minimum. Note, that the reference proves that a small $\eta<4$ is a valid gradient confusion constant with high probability only when the data are sampled from a unit sphere.
>
> **W3:** *... Current experiments are fairly limited (MNIST is the only dataset used and is easy to learn. Additionally, model architecture is also quite limited. Using other datasets e.g. CIFAR10/100 or running the experiments for networks with varying widths might be useful).*
>
> **A:** Thanks for the comment. We conducted some preliminary experiments on CIFAR-10 dataset for additional empirical evaluations. Please see the results and settings in the response to all reviewers above. Figures are shown in the pdf file attached to it. More extensive numerical evaluations are currently running and will be included in the revision. From what we can tell so far, the qualitative results we see on MNIST are also visible on CIFAR10, with the main difference that both GD and SGD are *both* much slower, as one expects.
>
>
>
> **Q1:** *There is no figure 2.*
>
> **A:** This is the figure on top of page 4; we will add a label to this figure in the revision.

---

> > ### Comment · Reviewer_FvUq · 2023-08-21
> >
> > I acknowledge and appreciate the authors' responses. I intend to keep my score.

---

### Official Review · Reviewer_eCMo · 2023-07-09

**Soundness:** 3 good
**Presentation:** 3 good
**Contribution:** 2 fair
**Rating:** 5
**Confidence:** 2

**Summary:**

This paper studies the convergence of SGD with large step size. It is shown that under some regularity conditions, SGD enjoys a fast linear convergence rate, both in expectation and with high probability. These results can be applied to show fast convergence of SGD for wide enough feed-forward networks.

**Strengths:**

The paper is well organized and easy to follow, and especially the authors have compared in detail the assumptions/results with those in the existing papers.

It is indeed an important question to study the convergence of SGD with large step sizes, and the results in this paper seem novel and interesting.

**Weaknesses:**

1. The main application of the results is for wide enough neural networks in the NTK regime, which seems restrictive.
2. It would be helpful if the authors can highlight the technical novelties.
3. It is discussed around line 40-42 that large batch sizes could be beneficial for generalization. However, the current results don't seem to have such implications. In particular, I don't think the regime studied in the current paper is a case of the "edge of stability" regime.
4. Is it possible to verify the proposed conditions for common neural network models, at least empirically? Otherwise it's not clear if the developed theory reflects what happens in practice.

**Questions:**

Please see the question above.

**Limitations:**

The authors have adequately addressed the limitations.

---

> ### Author Rebuttal · Authors · 2023-08-08
>
> We thank the reviewer for the constructive comments. We address the concerns as follows:
>
> **W1:** *The main application of the results is for wide enough neural networks in the NTK regime, which seems restrictive.*
>
> **A:** We would like to point out that the ”NTK regime“ setting is in line with much of the literature of optimization theories for neural networks.
> Nearly  all convergence results for GD/SGD for neural networks so far are in the NTK regime (or more precisely, require sufficiently large network width) (see, e.g., [1-5]).
> To the best of our knowledge, there is no general theory for convergence of GD/SGD outside of large width regimes.   The goal of our work is to improve the existing results on convergence of SGD by showing essentially the same rate of convergence as GD (something which has been empirically observed) and thus requires a setting where the existing analyses of GD exists.
> Furthermore, the large width regime should logically be the first to be explored before further progress can be made.
>
> [1] Allen-Zhu, Li, Song. A Convergence Theory for Deep Learning via
> Over-Parameterization. ICML. 2019.
>
> [2] Du, Zhai, Poczos, Singh. Gradient Descent Provably Optimizes
> Over-parameterized Neural Networks. ICLR, 2018.
>
> [3] Zou, Cao, Zhou, Gu. Gradient descent optimizes overparameterized deep ReLU networks. Machine Learning, 2020.
>
> [4] Oymak, Soltanolkotabi. Toward moderate overparameterization: Global convergence guarantees for training shallow neural networks. : IEEE Journal on Selected Areas in
> Information Theory, 2020.
>
> [5] Liu, Zhu, Belkin, Loss landscapes and optimization in over-parameterized non-linear systems and neural networks. ACHA, 2022.
>
> **W2:**  *It would be helpful if the authors can highlight the technical novelties.*
>
>
> **A:** Thanks for the suggestion. We will include a more comprehensive summary of contributions in the revision.
> From a high level viewpoint, we (1) introduced the aiming condition and (2) proved that sufficiently wide neural networks satisfy the aiming condition. Note that the width requirement we have is the same as that in establishing convergence of GD in prior work, e.g., [1]; hence, we are not imposing stronger assumptions.
> Second, we proved that aiming, interpolation, and quadratic growth conditions enable application of SGD with a large step size and fast linear rate.  Indeed, Section 1.2 explains in detail that the linear rate we obtain is much faster than existing convergence guarantees for SGD.
>
> From a technical viewpoint, we introduced a number of new analysis techniques. First, the proof that neural networks satisfy aiming and uniform aiming relied on some careful computations with nonlinear least squares and the transition to linearity phenomenon. Second, the proof of convergence of SGD relied on decomposing the iterate trajectories into    “tangent” and “orthogonal” components to the solution set $S$. The aiming condition ensured that the iterates make steady progress in orthogonal directions towards $S$ while uniform aiming controlled the tangent motion. Third, using this decomposition together with a novel martingale stopping time argument, we bounded the probability that the iterates escape the ball. We expect that such an orthogonal decomposition  with a stopping time argument to be useful more broadly.
>
>
>  [1] Liu, Zhu, Belkin, Loss landscapes and optimization in over-parameterized non-linear systems and neural networks, ACHA, 2022.
>
> **W3：** *It is discussed around line 40-42 that large batch sizes could be beneficial for generalization. However, the current results don't seem to have such implications. In particular, I don't think the regime studied in the current paper is a case of the "edge of stability" regime.*
>
>
> **A:** Thank you. This comment, made in passing, may have caused some confusion.
> We discuss generalization and the "edge of stability" as the motivations for understanding the training efficiency of SGD with large stepsize. We don't claim that our analysis implies better generalization or that it is in the  "edge of stability" regime. Instead, we focus on the optimization performance of SGD with large step size on the training loss only. As we have stated in Section 1.1 (outline of main results) and Section 1.2 (comparison to existing work), our goals are to improve the theoretically allowed stepsize range (especially to include the practically used large stepsizes), and to deduce faster convergence rate on the training loss. Apologies for the confusion; we will make this point clear in the revision.
>
> **W4** *Is it possible to verify the proposed conditions for common neural network models, at least empirically?*
>
> **A:** Thank you; this is a good question. The difficulty is that the aiming condition involves $proj_S(w)$---the nearest point of $w$ to the solution set $S$---and which we have no way of computing. Instead, we performed the following experiment on MNIST and CIFAR-10 datasets. We let SGD run and convergence to some point $\bar w$. Then we plot the quotients $\hat \theta_t\triangleq\langle \nabla L(w_t),w_t-\bar w\rangle/L(w_t)$ along the iterate path. Note that positivity of $\hat \theta_t$ suggests that aiming holds, but does not verify it exactly. Indeed, for MNIST we see that $\hat\theta_t>0.5$ while for CIFAR $\hat\theta_t>0.05$. See the results in the response to all reviewers above. Figures are shown in the pdf file attached to it.  We will include the empirical verification in the revision. We note that it may seem that the experiments verify the related quasar-convexity property, but this *not* the case because $\bar w$ is a *random point* that depends on the sample path taken by SGD.

---

> > ### Comment · Reviewer_eCMo · 2023-08-18
> > **Response to the authors**
> >
> > I thank the authors for the detailed response. I don't have further questions.

---

### Author Rebuttal · Authors · 2023-08-08

We provide additional experimental results here, requested by the reviewers. Please see the other rebuttals below, directly after each review.

**1: An estimate of aiming condition:** In principle, the aiming condition is difficult to verify, as
it involves $proj_S(w)$---the nearest point of $w$ to the solution set $S$---and which we have no way of computing. Instead, we replace $proj_S(w)$ by the last iterate $\bar{w}$ of an SGD run, and compute an estimate of the aiming condition coefficient by the quotients $\hat \theta_t\triangleq\langle \nabla L(w_t),w_t-\bar w\rangle/L(w_t)$ along the iterate path. Note that this quotient is not the same quotient as would appear in the definition of quasar-convexity because $\bar w$ is a random point that depends on the iterate path taken by SGD.



*Settings:*  We conduct the experiments on two datasets, MNIST and CIFAR-10. For MNIST, we trained a $3$-hidden layer fully-connected neural network (width$=1000$). First, we train the network until convergence (loss function value $< 10^{-4}$), and record the parameters of this trained network as the optimal solution $\bar{w}$.
For CIFAR-10, we train a two-layer CNN for $4000$ iterations $^\dagger$, and record the final parameter setting as $\bar{w}$. Then, for each experiment, we took a second run, and at each iteration $w_t$ we compute the estimate of the aiming condition coefficient $\hat \theta_t$ using the equation mentioned above.


*Results:* Figure 1 and Figure 2 in the attached pdf file show the plots of $\hat \theta_t$ against the iteration number. In both cases, we observe that the estimate $\hat \theta_t$ stays positive, which suggests that aiming holds.

We will include the empirical verification in the revision.


**2: Mini-batch SGD on CIFAR-10:** In this experiment, we show that, with the same number of iterations, mini-batch SGD has almost the same convergence behavior as the full batch GD on the CIFAR-10 dataset (same as in Figure 1 for MNIST).

*Setting:* We run mini-batch SGD with different batch sizes $b$ ($b=60, 120, 250, 500, 1000$), as well as full batch GD, on a two-layer CNN with CIFAR-10 dataset for $4000$ iterations $^\dagger$.

*Results:* Figure 3 in the attached pdf file shows the training loss curves of the mini-batch SGD and full batch GD. We observe that with the same number of iterations, mini-batch SGD has almost the same convergence behavior as the full batch GD.

$\dagger$: We didn't train this CNN until convergence, as the convergence is slow for CIFAR-10 on this simple CNN. We are currently running experiments on deeper networks so that a near-zero training loss could be obtained. We will include it in the revision.

---

### Decision · Program_Chairs · 2023-09-21

**Decision:**

Accept (poster)

**Comment:**

The authors have addressed the reviewers’ concerns. Please incorporate all the discussions and new experiments into the final version.